# THE COMPUTATIONAL COMPLEXITY OF POSITIVE NON-CLASHING TEACHING IN GRAPHS

**Robert Ganian, Liana Khazaliya & Mathis Rocton**
Algorithms and Complexity Group, TU Wien, Vienna, Austria
{rganian,lkhazaliya,mrocton}@ac.tuwien.ac.at

**Fionn Mc Inerney**
Telefónica Scientific Research, Barcelona, Spain
fmcinern@gmail.com

## ABSTRACT

We study the classical and parameterized complexity of computing the positive non-clashing teaching dimension of a set of concepts, that is, the smallest number of examples per concept required to successfully teach an intelligent learner under the considered, previously established model. For any class of concepts, it is known that this problem can be effortlessly transferred to the setting of balls in a graph $G$. We establish (1) the NP-hardness of the problem even when restricted to instances with positive non-clashing teaching dimension $k = 2$ and where all balls in the graph are present, (2) near-tight running time upper and lower bounds for the problem on general graphs, (3) fixed-parameter tractability when parameterized by the vertex integrity of $G$, and (4) a lower bound excluding fixed-parameter tractability when parameterized by the feedback vertex number and pathwidth of $G$, even when combined with $k$. Our results provide a nearly complete understanding of the complexity landscape of computing the positive non-clashing teaching dimension and answer open questions from the literature.

## 1 INTRODUCTION

While typical machine learning models task a learner with finding a concept $C$ from a concept class $\mathcal{C}$ based on an—often randomly drawn—sample, in *machine teaching* (specifically in its commonly considered batch variant) the learner is provided a set of examples by a teacher; crucially, here the examples can be selected in a way which allows the concept to be reconstructed from as few examples as possible. Machine teaching is a core topic in computational learning theory and has found applications in a variety of areas including robotics (Thomaz & Cakmak, 2009; Akgun et al., 2012), trustworthy AI (Mei & Zhu, 2015; Zhang et al., 2018), inverse reinforcement learning (Ho et al., 2016; Brown & Niekum, 2019), and education (Zhu, 2015; Chen et al., 2018; Zhu et al., 2018). While numerous models of machine teaching have been investigated to date (Shinohara & Miyano, 1991; Goldman & Kearns, 1995; Goldman & Mathias, 1996; Zilles et al., 2011; Gao et al., 2017; Mansouri et al., 2019; Telle et al., 2019), in this article we focus on the recently developed *positive non-clashing teaching model* (Kirkpatrick et al., 2019; Fallat et al., 2023).

In non-clashing teaching, given a finite binary concept class $\mathcal{C}$, for each pair $C_1$, $C_2$ of distinct concepts in $\mathcal{C}$, at least one example provided for at least one of $C_1$ or $C_2$ must not be consistent with the other concept. A key feature of non-clashing teaching is that it is the most efficient model (in terms of the number of required examples) satisfying the Goldman-Mathias collusion-avoidance criterion (Goldman & Mathias, 1996)—the "gold standard" for ensuring that the learner cannot cheat, *e.g.*, via a hidden communication channel with the teacher. Moreover, a teaching model is *positive* if the examples provided for each concept $C$ are required to be positively labeled for $C$. The restriction of teaching models to the positive setting is common and well-motivated from applications in, *e.g.*, recommendation systems (Schwab et al., 2000), computational biology (Wang et al., 2006; Yousef et al., 2008), and grammatical inference (Stolcke & Omohundro, 1994; Denis, 2001); see also the early works of Angluin (Angluin, 1980a;b). Non-clashing teaching has been proposed and studied

in the positive setting not only within the initial papers introducing the concept (Kirkpatrick et al., 2019; Fallat et al., 2023), but also in subsequent works (*e.g.*, (Chalopin et al., 2024)).

While positive non-clashing teaching has the potential to be highly efficient, realizing this potential requires us to solve the computational task of actually constructing a small set of examples for the given concepts. More precisely, one aims at computing (a witness for) the teaching dimension of a given concept class, *i.e.*, the minimum integer $k$ such that each concept is provided with at most $k$ examples that satisfy the conditions of the model. On the positive side, instead of considering various different types of concepts and examples, we can restrict our attention to the setting where each concept is a ball in some input-specified graph $G$, and the possible examples are vertices of $G$. Indeed, it is known that any finite binary concept class $\mathcal{C} \subseteq 2^V$ can be represented by a set $\mathcal{B}$ of balls in a graph $G$ as follows: $V(G) = V \cup \{x_C \mid C \in \mathcal{C}\}$, $x_C$ is adjacent to $x_{C'}$ for all $C, C' \in \mathcal{C}$, $x_C$ is adjacent to $v \in V$ if and only if $v \in C$, and $\mathcal{B} = \{B_1(x_C) \mid C \in \mathcal{C}\}$ (Chalopin et al., 2023; 2024).

On the negative side, the problem is computationally intractable, and remains so even in the restricted setting where *every* possible concept (*i.e.*, every ball in $G$) is present; to distinguish this case from the general one where not all concepts need to be present, we refer to it as *strict*. In particular, Chalopin, Chepoi, Mc Inerney, and Ratel (2024) recently carried out an initial complexity-theoretic investigation of computing the positive non-clashing teaching dimension in the strict setting. There, they established the NP-hardness of the problem for instances with large teaching dimension (even when restricted to the highly restricted class of split graphs), obtained runtime upper and lower bounds under the *Exponential Time Hypothesis* (Impagliazzo et al., 2001), and designed a so-called *fixed-parameter* algorithm for the problem when parameterized by the size of the vertex cover of $G$.

In this article, we significantly improve over each of these results, answer two open questions posed by the authors of the aforementioned work (Chalopin et al., 2024), and obtain a nearly complete understanding of the computational complexity of computing the positive non-clashing teaching dimension (in both the strict and non-strict settings).

**Contributions.** Let us refer to the problems of computing the positive non-clashing teaching dimension in the strict and non-strict settings as STRICT NON-CLASH and NON-CLASH, respectively. Formal definitions complementing the informal descriptions given above are provided in Section 2.

Our first result concerns the complexity of STRICT NON-CLASH on instances with constant positive non-clashing teaching dimension. The reductions of Chalopin, Chepoi, Mc Inerney, and Ratel (2024) only establish the NP-hardness of the problem for instances with large (*i.e.*, input-dependent) positive non-clashing teaching dimension $k$. In fact, as their first open question, the authors ask whether STRICT NON-CLASH is NP-hard or polynomial-time solvable when the sought-after dimension $k$ is a fixed constant; the question is not only theoretically interesting, but also highly relevant as instances of small teaching dimension are precisely the candidates for efficient teaching. We settle this by a highly non-trivial reduction (Theorem 1) which establishes that determining whether the positive non-clashing teaching dimension is at most 2 is NP-hard—and remains so even when restricted to the same class of *split graphs* where STRICT NON-CLASH was previously shown to be NP-hard (for large $k$) (Chalopin et al., 2024). We note that this result is, in a sense, best possible: determining whether an instance of STRICT NON-CLASH has a positive non-clashing teaching dimension of 1 is trivial as it is equivalent to testing whether $G$ is edgeless (Chalopin et al., 2024).

Next, we proceed to the running time bounds for solving the problem. Typically, the running time upper bounds are given by an exact algorithm, while the lower bounds are obtained from a suitable "tight" reduction under the Exponential Time Hypothesis (Impagliazzo et al., 2001). In the preceding work, Chalopin, Chepoi, Mc Inerney, and Ratel (2024) obtained algorithmic lower and upper bounds of $2^{o(n \cdot d)}$ and $2^{\mathcal{O}(n^2 \cdot d)}$, where $n$ and $d$ are the number of vertices and the diameter of $G$, respectively. From our reduction and a more careful algorithmic analysis, we obtain a lower bound of $2^{o(n \cdot d \cdot k)}$ and an upper bound of $2^{\mathcal{O}(n \cdot d \cdot k \cdot \log n)}$ (Theorem 4 and Proposition 5)—making the bounds almost tight, with just a logarithmic factor in the exponent separating the two.

While the aforementioned bounds apply to the problem in general, often one may only need to solve the problem on "well-structured" graphs. The more refined *parameterized complexity* paradigm (Downey & Fellows, 2013; Cygan et al., 2015) offers the perfect tools to analyze and identify precisely which structural properties of input graphs—usually captured by a suitable integer *parameter* $k$—allow us to circumvent its general intractability. The analog to the complex-

ity class P in the parameterized setting is FPT ("*fixed-parameter tractable*"), which characterizes parameterized problems solvable in $f(k) \cdot n^{\mathcal{O}(1)}$ time; intuitively, this means that the problem is solvable in uniformly polynomial time for each constant value of $k$. Parameterized complexity is well-established and has been successfully applied for non-clashing teaching (Chalopin et al., 2024) as well as in a variety of related subfields of learning theory (Downey et al., 1993; Li & Liang, 2018; Ganian & Korchemna, 2021; Ordyniak & Szeider, 2021; Brand et al., 2023; Eiben et al., 2023a;b).

In their previous work, Chalopin, Chepoi, Mc Inerney, and Ratel (2024) established the fixed-parameter tractability of STRICT NON-CLASH when parameterized by the *vertex cover number* of the input graph $G$—or, equivalently, the vertex deletion distance to a graph consisting only of isolated vertices. The drawback of that result is that the vertex cover number is a highly "restrictive" parameter, in the sense that it achieves low values only on rather simple graphs. This is also reflected in the open question posed in that article, which asked about the problem's complexity under other parameterizations. As our third contribution, we establish (in Theorem 17) the fixed-parameter tractability of NON-CLASH—*i.e.*, the more general task of computing the non-clashing teaching dimension when not all concepts (*i.e.*, balls) need to be present—parameterized by the *vertex integrity* of $G$. Vertex integrity is a well-studied graph parameter (Drange et al., 2016; Gima et al., 2022; Gima & Otachi, 2024; Hanaka et al., 2024) that essentially captures the vertex deletion distance to a graph consisting only of small connected components; it is known (and easily observed) to be a less restrictive parameterization than the vertex cover number (cf. Section 2), meaning that our result significantly pushes the boundaries of tractability even for the simpler strict variant of the problem.

The proof of Theorem 17 is highly non-trivial. In particular, while the algorithm itself is simple and merely uses a data reduction technique ("*kernelization*" (Cygan et al., 2015)) that iteratively removes certain parts of the instance, the crucial correctness proof underlying the result is very involved and relies on identifying a carefully defined set of "canonical" examples for our instances. The algorithm is also constructive, meaning that it can output a set of examples for the concepts as a witness.

As our final contribution, in Theorem 18 we complement Theorem 17 with a complexity-theoretic lower bound excluding fixed-parameter tractability under many other graph parameters previously considered in the literature, including *pathwidth*, *treewidth*, and the *feedback vertex number*—the latter two of which were explicitly mentioned in the aforementioned open question (Chalopin et al., 2024). We do so through a complex W[1]-*hardness reduction* (which can be seen as a parameterized analog to classical reductions used to establish NP-hardness) that excludes, under well-established complexity assumptions, NON-CLASH from being in FPT even when combining all the parameterizations mentioned in the previous sentence with the positive non-clashing teaching dimension $k$.

**Related Work.** It is known that NON-CLASH is significantly more challenging than the special case captured by STRICT NON-CLASH. For instance, the reduction of Kirkpatrick, Simon, and Zilles (2019, Subsection 7.1) establishes that NON-CLASH is NP-hard even if the task is to determine whether the positive non-clashing teaching dimension of the instance is 1; on the other hand, the analogous question for STRICT NON-CLASH is trivial as it simply requires determining whether the input graph is edgeless or not (Chalopin et al., 2024). In fact, unless P = NP, that reduction also rules out a polynomial-time 1.999-approximation algorithm. For clarity, note that while that reduction does not consider concepts that are balls in a graph, as mentioned earlier, *every* finite binary concept class can be easily transformed into a class of balls in a graph (Chalopin et al., 2023; 2024).

Apart from the computational questions resolved in this work, another prominent open question is whether the non-clashing teaching dimension is upper-bounded by the VC-dimension (Kirkpatrick et al., 2019; Fallat et al., 2023; Simon, 2023). It is known that the non-clashing teaching dimension (where one allows negative examples) can be significantly smaller than the positive variant, *e.g.*, balls in cycles have a non-clashing teaching dimension of 2, but their positive non-clashing teaching dimension is not bounded by any constant (Chalopin et al., 2024). It was also pointed out that balls in cacti or planar graphs could be good candidates for concept classes negatively answering this question (Chalopin et al., 2024). Further, Simon (2023) recently explored the relationship between non-clashing teaching and *recursive teaching* and identified the precise gap between the two notions.

Concept classes consisting of balls in a graph are a discrete analog of the geometric concept classes of balls in a Euclidean space which have been investigated in PAC-learning, *e.g.*, as part of the more general Dudley concept classes (Floyd, 1989; Ben-David & Litman, 1998). Apart from non-clashing teaching, they have also been explored for the closely related and well-studied sample compression

schemes (Chalopin et al., 2023) introduced by Littlestone and Warmuth (1986). As discussed in prior works (Kirkpatrick et al., 2019; Fallat et al., 2023; Chalopin et al., 2024), non-clashing teaching maps can be viewed as signed variants of representation maps for concept classes, a notion introduced to design unlabeled sample compression schemes for maximum concept classes (Kuzmin & Warmuth, 2007) (and subsequently the more general ample concept classes (Chalopin et al., 2022)).

## 2 PRELIMINARIES

We assume familiarity with graph terminology (Diestel, 2024). We only consider simple, finite, and undirected graphs. For an integer $n \geq 1$, we set $[n] := \{1, \ldots, n\}$. As we only consider finite binary concept classes which can be represented as balls in graphs (Chalopin et al., 2023; 2024), we introduce the terminology for positive non-clashing teaching directly in the setting of graphs.

POSITIVE NON-CLASHING TEACHING IN GRAPHS. Let $G$ be a graph. For an integer $r \geq 0$ and a vertex $v \in V(G)$, the *ball* $B_r(v)$ is the set of all vertices at distance at most $r$ from its *center* $v$. Let $\mathcal{B}$ be a set of balls of $G$. A *positive teaching map* $T$ for $\mathcal{B}$ is a mapping which assigns to each ball $B \in \mathcal{B}$ a *teaching set* $T(B) \subseteq B$, *i.e.*, a subset of the vertices of $B$. The *dimension* of $T$ is $\max_{B \in \mathcal{B}} |T(B)|$—in other words, the largest image of $T$. A positive teaching map $T$ is called *non-clashing* for $\mathcal{B}$ if for each pair of distinct balls $B_1, B_2 \in \mathcal{B}$, there exists a vertex $w \in T(B_1) \cup T(B_2)$ such that $w \notin B_1 \cap B_2$. Note that $w$ must lie in $B_1 \cup B_2$ by definition, and hence, this condition ensures that one of the balls has a teaching set which is not contained in the other ball. We say that $w$ *distinguishes* $B_1$ and $B_2$, or distinguishes $B_1$ from $B_2$ (or vice versa). If a teaching map is not non-clashing, we say that there is a *conflict* between any two balls for which there is no element distinguishing them. We now define our problems of interest:[1]

---

STRICT NON-CLASH
**Input:**      A graph $G$ and an integer $k$.
**Question:**  Is there a positive non-clashing teaching map for the set of all balls of $G$ with dimension at most $k$?

---

NON-CLASH
**Input:**      A graph $G$, a set $\mathcal{B}$ of balls of $G$, and an integer $k$.
**Question:**  Is there a positive non-clashing teaching map for $\mathcal{B}$ with dimension at most $k$?

---

We call a teaching map satisfying the conditions of the respective problem statement a *solution*. To avoid any confusion, we remark that the above definitions—as well as every result obtained in this article—concerns non-clashing teaching in the previously studied *positive* setting.

PARAMETERIZED COMPLEXITY. In parameterized complexity (Downey & Fellows, 2013; Cygan et al., 2015), the running-times of algorithms are studied with respect to a parameter $p \in \mathbb{N}$ and input size $n$. It is normally used for NP-hard problems, with the aim of finding a parameter describing a feature of the instance such that the combinatorial explosion is confined to this parameter. A parameterized problem is *fixed-parameter tractable* (FPT) if it can be solved by an algorithm running in time $f(p) \cdot n^{\mathcal{O}(1)}$, where $f$ is a computable function; these are *fixed-parameter algorithms*. Proving that a problem is W[1]-hard via a *parameterized reduction* from a W[1]-hard problem rules out the existence of a fixed-parameter algorithm under the well-established hypothesis that W[1] $\neq$ FPT.

STRICT NON-CLASH is known to be fixed-parameter tractable when parameterized by the *vertex cover number* of $G$, *i.e.*, the minimum integer $a$ such that there is a subset $X \subset V(G)$ of at most $a$ vertices where $G - X$ is an edgeless graph. In this article, we consider three parameters which are upper-bounded by the vertex cover number (or, more precisely, the vertex cover number plus one):

- the *vertex integrity* of $G$, which is the minimum integer $b$ such that there is a vertex subset $X \subset V(G)$ where for every connected component $H$ of $G - X$, $|V(H) \cup X| \leq b$;

- the *feedback vertex number* of $G$ (denoted by $\mathtt{fvs}(G)$), which is the minimum integer $c$ such that there is a vertex subset $X \subset V(G)$ where $G - X$ is acyclic;

---

[1]While we use decision variants, our algorithms are constructive and can output a teaching map as a witness.

- the *pathwidth* of $G$ (denoted by $\mathrm{pw}(G)$), which has a more involved definition based on the notion of *path decompositions*. However, for the purposes of this article it is sufficient to note the well-known facts (Downey & Fellows, 2013; Cygan et al., 2015) that deleting one vertex from each connected component of $G$ will decrease the pathwidth by at most one, and that a graph consisting of a disjoint union of paths and *subdivided caterpillars* (*i.e.*, graphs consisting of a central path with pendent paths attached to it) has pathwidth 2.

## 3 INTRACTABILITY AND RUNNING TIME LOWER BOUNDS

In this section, we establish that STRICT NON-CLASH is NP-hard even when $k = 2$, and thus, that it cannot be $1.499$-approximated in polynomial time unless $\mathsf{P} = \mathsf{NP}$. Recall that the former result is tight in the sense that STRICT NON-CLASH is trivial when $k = 1$ (Chalopin et al., 2024).

**Theorem 1.** STRICT NON-CLASH *is* NP-*hard even when restricted to split graphs with $k = 2$.*

We prove Theorem 1 via a polynomial-time reduction that, given an instance of 3-SAT, constructs an equivalent instance $(G, k)$ of STRICT NON-CLASH, where $G$ is a split graph and $k = 2$.

---

3-SAT
**Input:** A CNF formula over a set of clauses $\mathcal{C} = \{c_1, \ldots, c_m\}$ containing variables from $\mathcal{X} = \{x_1, \ldots, x_n\}$, where each clause has exactly 3 literals.
**Question:** Is there a variable assignment $\tau : \mathcal{X} \to \{\texttt{True}, \texttt{False}\}$ satisfying each clause in $\mathcal{C}$?

---

*Reduction.* Given an instance $\phi = (\mathcal{C}, \mathcal{X})$ of 3-SAT, we construct the graph $G$ as follows (see Figure 1b for an illustration).

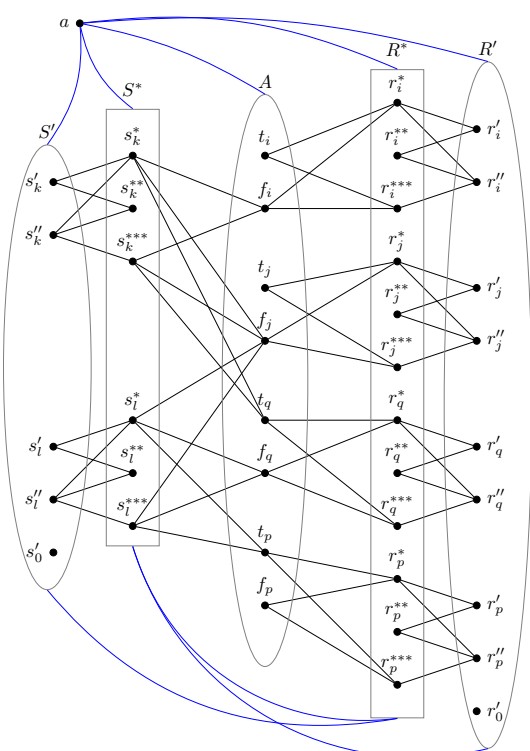

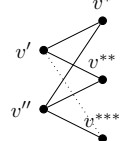

(a) The force-gadget. The dotted edge corresponds to the absence of that edge.

(b) An example of the graph $G$ obtained by applying our reduction on the 3-SAT instance with $\mathcal{X} = \{x_i, x_j, x_q, x_p\}$ and $\mathcal{C} = \{(x_i \lor x_j \lor \overline{x_q}), (x_j \lor x_q \lor \overline{x_p})\}$. Vertices in ovals form independent sets, while cliques are depicted by rectangles. Blue edges denote the existence of all possible edges between the two sets.

- First, for each $i \in [n]$, we create a pair of vertices $\{t_i, f_i\}$. We set $A := \{t_i, f_i\}_{i \in [n]}$.

- For each $i \in [n]$, we introduce a *variable force-gadget*, which consists of a set of vertices $\{r_i^*, r_i^{**}, r_i^{***}, r_i', r_i''\}$ and edges as depicted in Figure 1a.

- For each $i \in [n]$, we attach the variable force-gadget to the pair $\{t_i, f_i\}$ as shown in Figure 1b, by making both $r_i^*$ and $r_i^{***}$ adjacent to both $t_i$ and $f_i$. This gadget will guarantee that the corresponding assignment of the $i^{\text{th}}$ variable is well-defined. We set $R^* := \{r_i^*, r_i^{**}, r_i^{***}\}_{i \in [n]}$ and $R' := \{r_0'\} \cup \{r_i', r_i''\}_{i \in [n]}$, where $r_0'$ is a new vertex.

- Similarly, for each $k \in [m]$, we introduce a *clause force-gadget* on the set of new vertices $\{s_k^*, s_k^{**}, s_k^{***}, s_k', s_k''\}$ (as depicted in Figure 1a). This gadget corresponds to the clause $c_k$ of the instance $\phi$. We add adjacencies according to the appearance of literals in $c_k$, *i.e.*, if $x_i \in c_k$ for some $i \in [n]$, then we connect both $s_k^*$ and $s_k^{***}$ to $f_i$; and if $\overline{x_i} \in c_k$, then we connect both $s_k^*$ and $s_j^{***}$ to $t_i$. Intuitively, we connect the gadget to those vertices whose underlying assignments (True or False for $t_i$ and $f_i$, resp.) **do not** satisfy $c_j$, while the opposite assignments would (see Figure 1b). We set $S^* := \{s_k^*, s_k^{**}, s_k^{***}\}_{k \in [m]}$ and $S' := \{s_0'\} \cup \{s_k', s_k''\}_{k \in [m]}$, where $s_0'$ is a new vertex.

- We add all possible edges between (a) $S^*$ and $R^*$; (b) $R^*$ and $S'$; (c) $S^*$ and $R'$.

- We add all possible edges within $S^*$, and within $R^*$.

- Lastly, we add a special vertex $a$ and make it adjacent to all the other vertices of the graph.

This concludes the construction of $G$; given an instance $\phi = (\mathcal{C}, \mathcal{X})$ of 3-SAT, the reduction outputs the STRICT NON-CLASH instance $(G, 2)$. We now prove its correctness via the next two lemmas.

**Lemma 2.** *If $\phi$ is a YES-instance of 3-*SAT*, then $(G, 2)$ is a YES-instance of* STRICT NON-CLASH.

**Lemma 3.** *If $(G, 2)$ is a YES-instance of* STRICT NON-CLASH*, then $\phi$ is a YES-instance of 3-*SAT.

The proof of Theorem 1 then follows from Lemma 2 and Lemma 3. In particular, they prove that there is a polynomial-time reduction which transforms any instance of 3-SAT with $n$ variables and $m$ clauses into an equivalent instance $(G, 2)$ of STRICT NON-CLASH where $|V(G)| = \mathcal{O}(n + m)$ and $G$ is a split graph of diameter 2. The properties of this reduction also allow us to infer more precise algorithmic lower bounds. In particular, since an algorithm solving STRICT NON-CLASH in $2^{o(|V(G)| \cdot d \cdot k)}$ time would allow us to solve 3-SAT in $2^{o(n+m)}$ time:

**Theorem 4.** *Unless the Exponential Time Hypothesis fails, there is no algorithm solving* STRICT NON-CLASH *in time $2^{o(|V(G)| \cdot d \cdot k)}$, where $d$ and $k$ are the diameter of $G$ and the target positive non-clashing teaching dimension of the instance, respectively.*

We complement this lower bound with a refined upper bound for the more general NON-CLASH:

**Proposition 5.** NON-CLASH *can be solved in $2^{\mathcal{O}(|V(G)| \cdot d \cdot k \cdot \log |V(G)|)}$ time.*

## 4 FIXED-PARAMETER TRACTABILITY VIA VERTEX INTEGRITY

Given that NON-CLASH is NP-hard, it is natural to ask whether the problem can be solved efficiently on inputs exhibiting some well-defined structural properties. In this section, we establish the fixed-parameter tractability of NON-CLASH when parameterized by the vertex integrity of the input graph. Consider an instance $(G, \mathcal{B}, k)$ of NON-CLASH and let $p$ be the vertex integrity of $G$. As the first step, we invoke the known algorithm to compute a "witness" for the vertex integrity in time $p^{\mathcal{O}(p)}|V(G)|$ (Fellows & Stueckle, 1989), *i.e.*, a set $X \subset V(G)$ such that $|V(H) \cup X| \leq p$ for each connected component $H$ of $G - X$. Let $\mathcal{H}$ denote the set of connected components of $G - X$. To make use of the vertex integrity of $G$, we will partition the components of $\mathcal{H}$ into a parameter-bounded number of equivalence classes such that the elements belonging to the same class share some structural properties that will allow us to consider them, to some extent, interchangeable.

**Definition 1.** *Two subgraphs $H, H' \in \mathcal{H}$ are* twin-blocks *with respect to $\mathcal{B}$, denoted $H \sim_{\mathcal{B}} H'$, if there exists an isomorphism $\alpha_{H,H'}$ from $H$ to $H'$ with the following properties:*

- *for each $u \in V(H)$ and $v \in X$, $uv \in E(G)$ if and only if $\alpha_{H,H'}(u)v \in E(G)$, and*

- *for each $u \in V(H)$ and $r \in \mathbb{N}$, $B_r(u) \in \mathcal{B}$ if and only if $B_r(\alpha_{H,H'}(u)) \in \mathcal{B}$.*

Intuitively, $H \sim_{\mathcal{B}} H'$ if and only if there is a bijection $\alpha_{H,H'}$ between the vertices of the two subgraphs which preserves (1) edges inside $H$ and $H'$, (2) edges to $X$, and (3) the existence of balls in $\mathcal{B}$ centered at the vertices of $H$ and $H'$. We refer to $\alpha_{H,H'}$ as the *canonical isomorphism* between the two twin-blocks at hand, and if multiple choices of $\alpha$ exist, we choose and fix one arbitrarily; we drop the indices of $\alpha$ when the subgraphs are clear from the context. Observe that for any choice of $H$ and $H'$, $H \sim_{\mathcal{B}} H'$ can be tested in time at most $p^{\mathcal{O}(p)}$ by enumerating all possible choices of $\alpha$.

Clearly, $\sim_{\mathcal{B}}$ is an equivalence relation and we denote by $[H]_{\sim_{\mathcal{B}}}$ the equivalence class containing $H$. For $u \in V(H)$, we further define $[u]_{\sim_{\mathcal{B}}} = \{\alpha_{H,H'}(u) \mid H' \in [H]_{\sim_{\mathcal{B}}}\}$, and similarly for $B_r(u) \in \mathcal{B}$, $[B_r(u)]_{\sim_{\mathcal{B}}} = \{B_r(u') \in \mathcal{B} \mid u' \in [u]_{\sim_{\mathcal{B}}}\}$; intuitively, these refer to the sets of counterparts of $u$ and $B_r(u)$ in the equivalence class, respectively. For brevity, we overload the notation $\sim_{\mathcal{B}}$ and use $v \sim_{\mathcal{B}} w$ (or $B_r(v) \sim_{\mathcal{B}} B_r(w)$) to express that $v \in [w]_{\sim_{\mathcal{B}}}$ (or $B_r(v) \in [B_r(w)]_{\sim_{\mathcal{B}}}$, respectively).

**Observation 6.** *The number of equivalence classes on $\mathcal{H}$ defined by $\sim_{\mathcal{B}}$ is at most $2^{\mathcal{O}(p^3)}$.*

While the equivalence relation $\sim_{\mathcal{B}}$ is defined based on the input (in particular, $G$ and $\mathcal{B}$), our proof requires also considering a more refined equivalence relation based on the structure of a hypothetical positive non-clashing teaching map. Toward this, we use the following notion to capture how a hypothetical teaching set interacts with the balls centered in the components of $\mathcal{H}$.

**Definition 2.** *The* blueprint $S$ *of a teaching set $T(B)$ for a ball $B = B_r(u)$ centered in $H \in \mathcal{H}$ is a tuple $(S_X, S_H, S_f)$ composed of:*

1. *the set $S_X = T(B) \cap X$,*

2. *the set $S_H = T(B) \cap V(H)$,*

3. *the set $S_f = \{f_{H_0} \mid H_0 \in \mathcal{H}\}$ of functions, where for each $H_0$ the function $f_{H_0} : V(H_0) \to \{0, 1, 2\}$ specifies whether for a vertex $v \in V(H_0)$, the set $([v]_{\sim_{\mathcal{B}}} \cap T(B)) \setminus V(H)$ of counterparts of $v$ outside of $H$ has size 0, 1 or at least 2.*

Intuitively, the blueprint specifies how the teaching set for $B$ interacts with (1) the set $X$ and (2) the vertices inside $H$ itself; for the rest of the graph, the blueprint also counts how many "equivalent" vertices it contains from each equivalence class of $\mathcal{H}$, *but only up to* 2. At this point, it may not be clear why we do not differentiate between any size greater than 2; the reason is that if the actual size is 3 or more, there are superfluous elements in the teaching set, as we prove below. In fact, we prove a more general statement which holds regardless of whether vertices in $H$ are counted or not.

**Lemma 7.** *Let $u \in V(G)$, $B = B_r(u) \in \mathcal{B}$, and $T$ be a positive non-clashing teaching map for $\mathcal{B}$. Suppose there exist $H_0 \in \mathcal{H}$ and $v \in V(H_0)$ such that $|[v]_{\sim_{\mathcal{B}}} \cap T(B)| \geq 3$. Then, there exists $z$ in $[v]_{\sim_{\mathcal{B}}} \cap T(B)$ such that removing $z$ from $T(B)$ yields a positive non-clashing teaching map for $\mathcal{B}$.*

As a corollary of Lemma 7, we can obtain an upper bound on the positive non-clashing teaching dimension of our instance, which will be useful later in this section.

**Corollary 8.** *Let $G$ be a graph with vertex integrity $p$ and $\mathcal{B}$ an arbitrary set of balls of $G$. Then, the positive non-clashing teaching dimension of $\mathcal{B}$ is upper-bounded by a function $s(p) = 2^{\mathcal{O}(p^3)}$.*

Returning to the notion of blueprints defined earlier, we can now formalize a refined notion of equivalence that also takes into account the behavior of a hypothetical solution.

**Definition 3.** *Given $B = B_r(u)$ centered in $H$, and $H' \sim_{\mathcal{B}} H$, we say that $B$ and $B' = B_r(\alpha(u))$ share the same blueprint if, for $(S_X, S_H, (f_{H_0})_{H_0 \in \mathcal{H}})$ and $(S'_X, S'_H, (f'_{H_0})_{H_0 \in \mathcal{H}})$, the respective blueprints of $B$ and $B'$, the following hold: (1) $S_X = S'_X$, (2) $\forall v \in V(H), v \in S_H \Leftrightarrow \alpha(v) \in S'_H$, and (3) $\forall H_0 \in \mathcal{H}, f_{H_0} = f'_{H_0}$.*

**Definition 4.** *Given a positive non-clashing teaching map $T$ for $\mathcal{B}$, we say that two components $H, H' \in \mathcal{H}$ are perfectly-equivalent twin-blocks (or simply perfectly equivalent) with respect to $T$ on $\mathcal{B}$, denoted $H \equiv_{\mathcal{B}}^{T} H'$, if $H \sim_{\mathcal{B}} H'$ and, for all $u \in V(H)$, $r \in \mathbb{N}$, and $B_r(u) \in \mathcal{B}$, it holds that $B_r(u)$ and $B_r(\alpha(u))$ share the same blueprint. Note that $\equiv_{\mathcal{B}}^{T}$ is an equivalence relation.*

Crucially, we can also bound the number of equivalence classes for this refined equivalence defined w.r.t. a hypothetical teaching map. We denote the equivalence class of $H$ by $[H]_{\equiv_{\mathcal{B}}^{T}}$.

**Observation 9.** *The number of equivalence classes on $\mathcal{H}$ defined by $\equiv_{\mathcal{B}}^{T}$ is at most $2^{2^{\mathcal{O}(p^3)}}$.*

With this refined equivalence in hand, we proceed to the second milestone required for our algorithm: a structural result establishing the existence of "well-behaved teaching maps". Roughly speaking, by "well-behaved" we mean teaching maps which only use examples (*i.e.*, vertices) from $X$ and a bounded number of "core" components in $\mathcal{H}$, with some controlled exceptions. Toward formalizing this, let us fix an arbitrary total ordering $\prec$ on the components in $\mathcal{H}$.

**Definition 5.** *For $B \in \mathcal{B}$ and $x \in T(B) \cap V(H)$ for some $H \in \mathcal{H}$, the $(B,x)$-core $\mathcal{K}_{(B,x)}$ contains the first $s(p)+1$ elements w.r.t. $\prec$ of $\{H' \in [H]_{\sim_{\mathcal{B}}} \mid \exists B' \in [B]_{\equiv_{\mathcal{B}}^T}, \alpha(x) \in T(B') \cap V(H')\}$. The $B$-core is the union of the $(B,x)$-cores for all such $x$, and the core $\mathcal{K}$ is the union for all $B \in \mathcal{B}$ of the $B$-cores.*

**Definition 6.** *A component $H \in \mathcal{H}$ is in the 1-extended core $\mathcal{K}_1$ if $H \in \mathcal{K}$ or there exists $B \in \mathcal{B}$ and $x \in V(H)$ such that (1) $B$ is centered in $\mathcal{K}$ or in $X$, and (2) $x \in T(B)$. Going one step further, a component $H \in \mathcal{H}$ is in the 2-extended core $\mathcal{K}_2$ if $H \in \mathcal{K}_1$ or there exists $B \in \mathcal{B}$ and $x \in V(H)$ such that (1) $B$ is centered in $\mathcal{K}_1$ or in $X$, and (2) $x \in T(B)$.*

For $A \in \{\mathcal{K}, \mathcal{K}_1, \mathcal{K}_2\}$, note that $A$ is a subset of $\mathcal{H}$, but by slight abuse of notation, we will write $x \in V(A)$ for $x \in V(G)$ to denote $\exists H \in A$ with $x \in V(H)$ when not leading to any ambiguity. It is not difficult to see that the definitions yield the following.

**Observation 10.** *The number of components in $\mathcal{K}_2$ is upper-bounded by $c(p) = 2^{2^{\mathcal{O}(p^3)}}$.*

We can now formalize the notion of "well-behaved" teaching maps via the property of compactness.

**Definition 7.** *A teaching map $T$ is compact if, for any $H \in \mathcal{H} \setminus \mathcal{K}_2$ and any $x \in V(H)$, each ball whose teaching set contains $x$ is centered in $H$.*

Our next aim is to prove in Lemma 13 that we can safely restrict our attention to computing compact teaching maps only. Before we can establish that result, we first prove two auxiliary statements.

**Lemma 11.** *Let $T$ be a positive non-clashing teaching map for $\mathcal{B}$. Let $B \in \mathcal{B}$ be centered in $H_B \in \mathcal{H} \setminus \mathcal{K}_1$ with $x \in T(B) \cap V(H)$ for some $H \in \mathcal{H} \setminus H_B$ and $H_i \in \mathcal{K}_{(B,x)}$. By definition, there exists $x_i \in [x]_{\sim_{\mathcal{B}}} \cap V(H_i)$. If replacing $x$ by $x_i$ in $T(B)$ creates a conflict, then there exists $z_i \in T(B) \cap V(H_i)$.*

**Lemma 12.** *Let $T$ be an optimal positive non-clashing teaching map for $\mathcal{B}$, with $B \in \mathcal{B}$ centered in $H_B \in \mathcal{H} \setminus \mathcal{K}_1$. For $x \in T(B) \cap V(H)$ for some $H \in \mathcal{H} \setminus H_B$, if $|\mathcal{K}_{(B,x)}| = s(p) + 1$, then there are $H_i \in \mathcal{K}_{(B,x)}$ and $x_i \in [x]_{\sim_{\mathcal{B}}} \cap V(H_i)$ such that replacing $x$ by $x_i$ in $T(B)$ creates no conflict.*

**Lemma 13.** *If $\mathcal{B}$ has positive non-clashing teaching dimension $k$, then it also admits a compact positive non-clashing teaching map of dimension $k$.*

We now proceed to the crux of our algorithm: the proof that one can reduce the size of the instance $(G, \mathcal{B})$ without changing its positive non-clashing teaching dimension (formalized in Lemma 15). Toward this, it will be useful to focus on how the balls in $\mathcal{B}$ interact with certain subgraphs of $G$.

**Definition 8.** *Let $G'$ be an induced subgraph of $G$. Then, $\mathcal{B}$ induces the set $\mathcal{B}'$ of vertex sets w.r.t. $G'$, where $\mathcal{B}' = \{B_r(u) \cap V(G') \mid B_r(u) \in \mathcal{B} \wedge u \in V(G')\}$.*

We note that $\mathcal{B}'$ need not necessarily be a set of balls in $G'$ itself: for instance, it may well happen that for some $B \in \mathcal{B}'$ the vertex set $B \cap V(G')$ is not even connected. However, under some conditions on $G'$ that we will be able to guarantee, $\mathcal{B}'$ is, in fact, a set of balls in $G'$.

**Lemma 14.** *Let $G$ be a graph, $\mathcal{B}$ a set of balls in $G$, and $G'$ an induced subgraph of $G$. If $V(G')$ contains $X$ and, for every $H \in \mathcal{H}$, either (1) $V(H) \subseteq V(G')$ or (2) $V(H) \cap V(G') = \emptyset \wedge \exists H', H'' \in [H]_{\sim_{\mathcal{B}}}, H' \neq H'', V(H') \cup V(H'') \subseteq V(G')$ holds, then the set $\mathcal{B}'$ induced by $\mathcal{B}$ w.r.t. $G'$ is a set of balls in $G'$. Moreover, there is a bijection between balls in $\mathcal{B}'$ and balls in $\mathcal{B}$ centered in $G'$.*

We define the *reduced graph $G'$ of $G$* as the graph obtained from $G$ by removing all but $f(p) = c(p) + \binom{p \cdot c(p) + p}{s(p)}^{b(p)} + 1$ twin-blocks from each large class of $\sim_{\mathcal{B}}$, where $b(p) = \mathcal{O}(p^3)$ is the maximum number of distinct balls centered in any component of $\mathcal{H}$. Let $\mathcal{B}'$ be the set induced by $\mathcal{B}$ on $G'$ according to Definition 8. By Lemma 14, $\mathcal{B}'$ is a set of balls in $G'$. The size of $G'$ is at most $g(p) = p + f(p) \cdot 2^{\mathcal{O}(p^3)} = 2^{2^{\mathcal{O}(p^3)}}$. Thus, an optimal positive non-clashing teaching map for $\mathcal{B}'$ can be computed (*e.g.*, by brute force) in time that depends only on $p$. We now aim at proving that $\mathcal{B}'$ is equivalent to $\mathcal{B}$, *i.e.*, that the optimal positive non-clashing teaching dimension is the same for both.

**Lemma 15.** *Suppose that there is a solution for $(G, \mathcal{B}, k)$. Then, there exists a solution $T'$ for the reduced instance $(G', \mathcal{B}', k)$.*

The following lemma establishes that a solution for the reduced instance can be lifted to one for the original instance; note that this also ensures that our algorithm will be constructive.

**Lemma 16.** *Let $k \in \mathbb{N}$ and $T'$ be a compact solution for the instance $(G', \mathcal{B}', k)$. We can construct a compact solution $T$ for $(G, \mathcal{B}, k)$ in $2^{2^{\mathcal{O}(p^3)}} \cdot |V(G)|^{\mathcal{O}(1)}$ time.*

Finally, we obtain our main parameterized tractability result by combining the previous ingredients. In particular, in the proof, we construct $(G', \mathcal{B}', k)$, solve the problem there, and argue correctness.

**Theorem 17.** Non-Clash *is* FPT *parameterized by the vertex integrity of the input graph $G$.*

## 5 Hardness for Classical Structural Parameterizations

In this section, we prove that Non-Clash is intractable w.r.t. the feedback vertex number and pathwidth; the same then holds for generalizations of these measures like treewidth or clique-width.

**Theorem 18.** Non-Clash *is* W[1]-*hard parameterized by* $\texttt{fvs}(G) + \texttt{pw}(G) + k$.

We establish Theorem 18 by describing a parameterized reduction below and proving its correctness in Lemma 19. We reduce from a problem called NAE-Integer-3-SAT, which is W[1]-hard parameterized by the number of variables (Bringmann et al., 2016), and is defined as follows.

---

NAE-Integer-3-SAT
**Input:** A set of clauses $\mathcal{C}$ over variables $\mathcal{X}$, and an integer $d$. Any clause $c \in \mathcal{C}$ has the form $x \leq c_x$, $y \leq c_y$, and $z \leq c_z$, where $c_x, c_y, c_z \in \{1, \ldots, d\}$.
**Question:** Is there a variable assignment $\mathcal{X} \rightarrow \{1, \ldots, d\}$ such that for each clause, either 1 or 2 of its three inequalities are satisfied?

---

*The Reduction.* Given an instance $\phi$ of NAE-Integer-3-SAT, we construct an instance $(G, \mathcal{B}, k)$ of Non-Clash in polynomial time as follows, starting with the graph $G$.

- For each variable $x \in \mathcal{X}$, make a (variable) path $P^x := (x_1, \ldots, x_d)$ of order $d$.

- For each clause $c \in \mathcal{C}$, make two vertices $c$ and $c'$ and, w.l.o.g., suppose that the clause $c$ contains the variables $x$, $y$, and $z$, and that $c_x \leq c_y \leq c_z$. Connect the vertex $c$ to $x_d$, $y_d$, and $z_d$ by three distinct paths of lengths $4d$, $4d + c_y - c_x$, and $4d + c_z - c_x$, respectively.[2] These long paths ensure that there are no unwanted shortcuts in $G$ and that, for $r_c = 5d - c_x - 1$, the ball $B_{r_c}(c)$ contains all of $P^x$, $P^y$, and $P^z$ except for their first $c_x$, $c_y$, and $c_z$ vertices (whose respective indices correspond to the respective variable values that satisfy the clause $c$), respectively. Similarly, connect the vertex $c'$ to $x_1$, $y_1$, and $z_1$ by three distinct paths of lengths $4d + c_z - c_x$, $4d + c_z - c_y$, and $4d$, respectively. Analogously, this ensures that, for $r'_{c'} = 4d + c_z - 1$, the ball $B_{r'_{c'}}(c')$ contains the opposite vertices to $B_{r_c}(c)$ in $P^x$, $P^y$, and $P^z$ (whose respective indices correspond to the respective variable values that do not satisfy the clause $c$), while ensuring that no unwanted shortcuts exist in $G$.

- For each clause $c \in \mathcal{C}$ and each variable $q \in \mathcal{X}$ such that $c$ does not contain $q$, in $G$, connect the vertices $c$ and $c'$ to $q_1$ by distinct paths of length $3d$. These paths ensure that the balls $B_{r_c}(c)$ and $B_{r'_{c'}}(c')$ described above contain every other variable path completely, while ensuring that no unwanted shortcuts exist in $G$.

- Let $S$ be the set of all the vertices that currently exist in $G$. For each $x \in \mathcal{X}$, in $G$, make a vertex $f_x$ and connect it to each vertex in $S$ except those in $P^x$ via a distinct path of length $6d$. Finally, for all $x, y \in \mathcal{X}$, make $f_x$ adjacent to $f_y$. This ensures that, for each $x \in \mathcal{X}$, the ball $B_{6d}(f_x)$ contains every vertex in $G$ except for those in $P^x$, while ensuring that no unwanted shortcuts exist in $G$. This completes the construction of $G$ (see Figure 2).

Set $k := |\mathcal{X}|$. Let $\mathcal{B}$ consist of $V(G)$, $B_{6d}(f_x)$ for all $x \in \mathcal{X}$, and $B_{r_c}(c)$ and $B_{r'_{c'}}(c')$ for all $c \in \mathcal{C}$.

---

[2]"Connect two vertices $u, v$ by a path of length $p$" means to make $u$ and $v$ the endpoints of a path of length $p$.

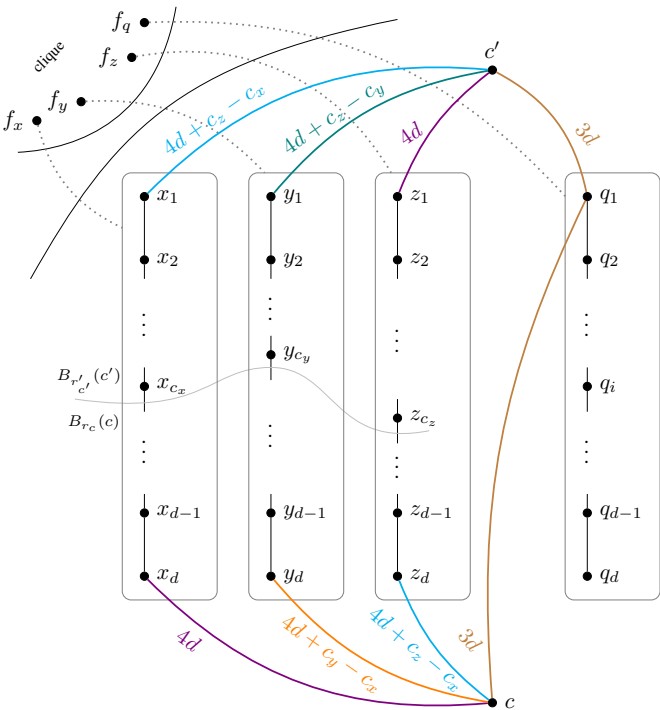

Figure 2: Colorful edges denote paths of the depicted lengths. Two black curves separate the clique from the rest of the graph. Each dotted edge shows that a vertex of the clique is adjacent to all the vertices of the graph below the separating curves except those in the adjacent rectangle. The gray curve gives an intuition of which vertices of $P_x$, $P_y$, and $P_z$ are contained in $B_{r_c}(c)$ and $B_{r'_{c'}}(c')$.

*Correctness of the Reduction.* Suppose, given an instance $\phi$ of NAE-INTEGER-3-SAT, that the reduction from the subsection above returns $(G, \mathcal{B}, k)$ as an instance of NON-CLASH.

**Lemma 19.** *$\phi$ is a YES-instance of* NAE-INTEGER-3-SAT *if and only if $(G, \mathcal{B}, k)$ is a YES-instance of* NON-CLASH.

*Proof of Theorem 18.* Lemma 19 establishes the correctness of the polynomial-time reduction from the beginning of Section 5. To complete the proof, it remains to show that $\mathtt{fvs}(G) + \mathtt{pw}(G) + k$ is bounded above by a function of $|\mathcal{X}|$. This clearly holds for $k$ which is, by definition, $|\mathcal{X}|$. Deleting from $G$ the vertices $x_1$, $x_d$, and $f_x$ for all $x \in \mathcal{X}$ results in an acyclic graph $G'$; in particular $G$ has a feedback vertex set of size $3|\mathcal{X}|$. To establish a bound on $\mathtt{pw}(G)$, it now suffices to show that $G'$ also has bounded pathwidth. Note that $G'$ consists of a set of connected components, each of which is either a subdivided caterpillar (this is what remains of each component containing a variable path) or a vertex (of the form $c$ or $c'$) with multiple pendent subdivided caterpillars and (simple) paths. Since deleting one further vertex from each connected component may only reduce the pathwidth by 1 and we need a single such deletion operation to reach a graph class of constant pathwidth (see Section 2), we also obtain that $\mathtt{pw}(G)$ is bounded by a function of $|\mathcal{X}|$. □

## 6 CONCLUDING REMARKS

Our computational upper and lower bounds provide a near-comprehensive understanding of the complexity of computing the positive non-clashing teaching dimension. Apart from our contributions to the previously studied strict setting, we consider it notable that our work is the first to also tackle the complexity of non-clashing teaching in the non-strict setting—*i.e.*, the more general (and arguably more natural) case where not all possible concepts are present. One open question highlighted by our work concerns the tiny remaining gap between the algorithmic lower and upper bounds obtained in Theorem 4 and Proposition 5. More general directions for future work are to perform a similar complexity analysis in the non-positive setting and to consider approximation algorithms.

## ACKNOWLEDGEMENTS

This work was funded by the Austrian Science Fund (FWF) [10.55776/Y1329 and 10.55776/COE12], the WWTF Vienna Science and Technology Fund (Project 10.47379/ICT22029), the ⬛ European Union's Horizon 2020 research and innovation COFUND programme Log-iCS@TUWien (grant agreement No 101034440), the Spanish Ministry of Economic Affairs and Digital Transformation and the European Union-NextGenerationEU through the project 6G-RIEMANN (TSI-063000-2021-147), and the Smart Networks and Services Joint Undertaking (SNS JU) under the European Union's Horizon Europe and innovation programme under Grant Agreement No. 101139067 (ELASTIC). Views and opinions expressed are however those of the authors only and do not necessarily reflect those of the European Union (EU). Neither the EU nor the granting authority can be held responsible for them.

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
