- which maps **YES**-instances to **YES**-instances and **NO**-instances to **NO**-instances,

- is computable in time $f(p) \cdot n^{\mathcal{O}(1)}$, where $f$ is a computable function, and

- where the parameter of the output instance can be upper-bounded by some function of the parameter of the input instance.

---

[1]While we use decision variants, our algorithms are constructive and can output a teaching map as a witness.

STRICT NON-CLASH is known to be fixed-parameter tractable when parameterized by the *vertex cover number* of $G$, *i.e.*, the minimum integer $a$ such that there is a subset $X \subset V(G)$ of at most $a$ vertices where $G - X$ is an edgeless graph. In this article, we consider three parameters which are upper-bounded by the vertex cover number (or, more precisely, the vertex cover number plus one):

- the *vertex integrity* of $G$, which is the minimum integer $b$ such that there is a vertex subset $X \subset V(G)$ where for every connected component $H$ of $G - X$, $|V(H) \cup X| \leq b$;

- the *feedback vertex number* of $G$ (denoted by $\mathtt{fvs}(G)$), which is the minimum integer $c$ such that there is a vertex subset $X \subset V(G)$ where $G - X$ is acyclic;

- the *pathwidth* of $G$ (denoted by $\mathtt{pw}(G)$), which has a more involved definition based on the notion of *path decompositions*. However, for the purposes of this article it is sufficient to note the well-known facts (Downey & Fellows, 2013; Cygan et al., 2015) that deleting one vertex from each connected component of $G$ will decrease the pathwidth by at most one, and that a graph consisting of a disjoint union of paths and *subdivided caterpillars* (*i.e.*, graphs consisting of a central path with pendent paths attached to it) has pathwidth 2.

## 3 INTRACTABILITY AND RUNNING TIME LOWER BOUNDS

In this section, we establish the NP-hardness of STRICT NON-CLASH when $k = 2$, and thus, that it cannot be $1.499$-approximated in polynomial time unless $\mathsf{P} = \mathsf{NP}$; in fact, our results hold even when the graphs belong to the class of *split graphs*, *i.e.*, graphs which can be partitioned into an independent set and a clique. Recall that the former result is tight in the sense that STRICT NON-CLASH is trivial when $k = 1$ (Chalopin et al., 2024). We formalize the result below.

**Theorem 1.** STRICT NON-CLASH *is* NP-*hard even when restricted to split graphs with $k = 2$.*

We prove Theorem 1 via a polynomial-time reduction that, given an instance of 3-SAT, constructs an equivalent instance $(G, k)$ of STRICT NON-CLASH, where $G$ is a split graph and $k = 2$.

---

**3-SAT**
**Input:** A CNF formula over a set of clauses $\mathcal{C} = \{c_1, \ldots, c_m\}$ containing variables from $\mathcal{X} = \{x_1, \ldots, x_n\}$, where each clause has exactly 3 literals.
**Question:** Is there a variable assignment $\tau : \mathcal{X} \to \{\texttt{True}, \texttt{False}\}$ satisfying each clause in $\mathcal{C}$?

---

*Reduction.* Given an instance $\phi = (\mathcal{C}, \mathcal{X})$ of 3-SAT, we construct the graph $G$ as follows (see Figure 1b for an illustration).

- First, for each $i \in [n]$, we create a pair of vertices $\{t_i, f_i\}$. We set $A := \{t_i, f_i\}_{i \in [n]}$.

- For each $i \in [n]$, we introduce a *variable force-gadget*, which consists of a set of vertices $\{r_i^*, r_i^{**}, r_i^{***}, r_i', r_i''\}$ and edges as depicted in Figure 1a.

- For each $i \in [n]$, we attach the variable force-gadget to the pair $\{t_i, f_i\}$ as shown in Figure 1b, by making both $r_i^*$ and $r_i^{***}$ adjacent to both $t_i$ and $f_i$. This gadget will guarantee that the corresponding assignment of the $i^{\text{th}}$ variable is well-defined. We set $R^* := \{r_i^*, r_i^{**}, r_i^{***}\}_{i \in [n]}$ and $R' := \{r_0'\} \cup \{r_i', r_i''\}_{i \in [n]}$, where $r_0'$ is a new vertex.

- Similarly, for each $k \in [m]$, we introduce a *clause force-gadget* on the set of new vertices $\{s_k^*, s_k^{**}, s_k^{***}, s_k', s_k''\}$ (as depicted in Figure 1a). This gadget corresponds to the clause $c_k$ of the instance $\phi$. We add adjacencies according to the appearance of literals in $c_k$, *i.e.*, if $x_i \in c_k$ for some $i \in [n]$, then we connect both $s_k^*$ and $s_k^{***}$ to $f_i$; and if $\overline{x_i} \in c_k$, then we connect both $s_k^*$ and $s_j^{***}$ to $t_i$. Intuitively, we connect the gadget to those vertices whose underlying assignments (True or False for $t_i$ and $f_i$, resp.) **do not** satisfy $c_j$, while the opposite assignments would (see Figure 1b). We set $S^* := \{s_k^*, s_k^{**}, s_k^{***}\}_{k \in [m]}$ and $S' := \{s_0'\} \cup \{s_k', s_k''\}_{k \in [m]}$, where $s_0'$ is a new vertex.

- We add all possible edges between (a) $S^*$ and $R^*$; (b) $R^*$ and $S'$; (c) $S^*$ and $R'$.

- We add all possible edges within $S^*$, and within $R^*$.

- Lastly, we add a special vertex $a$ and make it adjacent to all the other vertices of the graph.

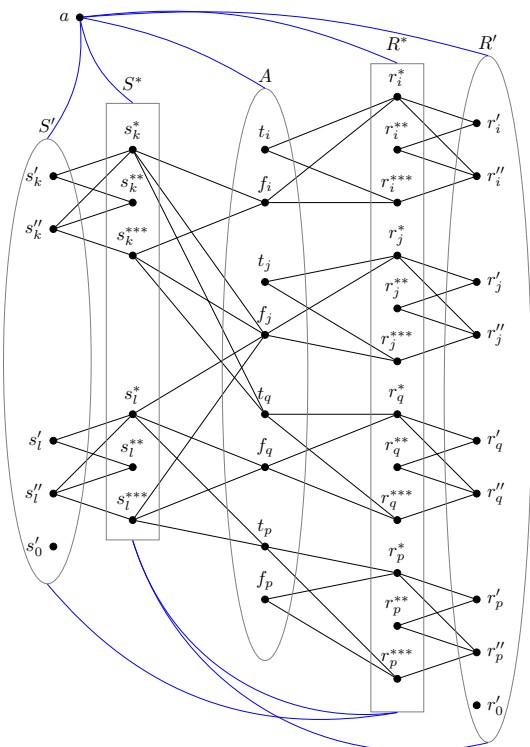

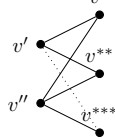

(a) The force-gadget. The dotted edge corresponds to the absence of that edge.

(b) An example of the graph $G$ obtained by applying our reduction on the 3-SAT instance with $\mathcal{X} = \{x_i, x_j, x_q, x_p\}$ and $\mathcal{C} = \{(x_i \vee x_j \vee \overline{x_q}), (x_j \vee x_q \vee \overline{x_p})\}$. Vertices in ovals form independent sets, while cliques are depicted by rectangles. Blue edges denote the existence of all possible edges between the two sets.

This concludes the construction of $G$; given an instance $\phi = (\mathcal{C}, \mathcal{X})$ of 3-SAT, the reduction outputs the STRICT NON-CLASH instance $(G, 2)$. We now prove its correctness via the next two lemmas.

**Lemma 2.** *If $\phi$ is a YES-instance of 3-SAT, then $(G, 2)$ is a YES-instance of* STRICT NON-CLASH.

*Proof.* As $G$ contains a universal vertex $a$, its diameter is 2, and thus, any ball of radius at least 2 contains $V(G)$. Hence, it is sufficient to consider balls of radius at most 2. Let $\tau : \mathcal{X} \rightarrow \{\text{True}, \text{False}\}$ be an assignment of the variables satisfying the given 3-SAT formula $\phi$. Let us define a positive teaching map $T$ of dimension 2 for the set $\mathcal{B}$ of all the balls in $G$ as shown in Table 1. It remains to prove that $T$ is non-clashing for $\mathcal{B}$.

To this end, we will refer to Table 2 which intuitively describes which vertices are used to distinguish each pair of balls of radius 1. For any pair $u, z \in V(G)$, the corresponding entry in the table contains either $v$ or a given vertex $w \in V(G)$. If it contains a vertex $w \in V(G)$, observe that $w \in T(B_1(u)) \cup T(B_1(z))$ by Table 1, and $w \notin B_1(u) \cap B_1(z)$. In the remaining cases, *i.e.*, (1) $s_k^*, r_i^*$; (2) $s_k^*, s_k^{**}$; (3) $s_k^*, s_k'$; (4) $s_k^*, s_k''$, we need to make it explicit which vertex $v$ stands for. For pairs (2) $s_k^*, s_k^{**}$; (3) $s_k^*, s_k'$; (4) $s_k^*, s_k''$, $v$ can be any vertex in $N(s_k^*) \cap A$ (by construction, there are 3 such vertices) since the neighborhoods of each of $s_k^{**}, s_k'$, and $s_k''$ do not intersect $A$.

For the last type of pair *(1)*, we will use the fact that $\tau$ is a satisfying assignment. For $k \in [m]$ and $i \in [n]$, assume w.l.o.g. that $c_k$ is satisfied by the assignment $\tau(x_i) = \text{True}$. Then, $T(B_1(s_k^*)) = \{s_k', f_i\}$ and $T(B_1(r_i^*)) = \{r_i', t_i\}$. With such an assignment of teaching sets, for any pair of type $s_k^*, r_i^*$ we have that $t_i \in T(B_1(s_k^*)) \cup T(B_1(r_i^*)) = \{s_k', f_i\} \cup \{r_i', t_i\}$, and $t_i \notin B_1(s_k^*) \cap B_1(r_i^*)$.

Finally, we check that the teaching set for $T(V(G))$ distinguishes $V(G)$ from all the other balls in $G$. According to Table 1, for $i, j \in [n]$ and $i \neq j$, $T(V(G)) = \{t_i, t_j\}$ and there is no $k \in [m]$ such that both $x_i$ and $x_j$ appear in $c_k$. The last condition guarantees us that, for any $u \in V(G) \setminus \{a\}$, there

| | |
|---|---|
| for $k \in [m]$ | $T(B_1(s_k^*)) = \{s_k', v\}$ and $c_k$ is satisfied by $\tau(x_i)$ for some $i \in [n]$, where |
| | $\quad$ if $\tau(x_i) = \text{True}$, then $v = f_i \qquad (f_i \in B_1(s_k^*) \cap A$ since $x_i \in c_k)$ |
| | $\quad$ if $\tau(x_i) = \text{False}$, then $v = t_i \qquad (t_i \in B_1(s_k^*) \cap A$ since $\overline{x_i} \in c_k)$ |
| | $T(B_1(s_k^{**})) = \{s_k', r_0'\} \quad T(B_1(s_k^{***})) = \{s_k'', r_0'\}$ |
| | $T(B_1(s_k')) = \{s_k', s_k^{**}\} \quad T(B_1(s_k'')) = \{s_k'', s_k^{***}\} \quad T(B_1(s_0')) = \{s_0', a\}$ |
| for $i \in [n]$ | $T(B_1(r_i^*)) = \{r_i', v\}$, where |
| | $\quad$ if $\tau(x_i) = \text{True}$, then $v = t_i$ |
| | $\quad$ if $\tau(f_i) = \text{False}$, then $v = f_i$ |
| | $T(B_1(r_i^{**})) = \{r_i', s_0'\} \quad T(B_1(r_i^{***})) = \{r_i'', s_0'\}$ |
| | $T(B_1(r_i')) = \{r_i', r_i^{**}\} \quad T(B_1(r_i'')) = \{r_i'', r_i^{***}\} \quad T(B_1(r_0')) = \{r_0', a\}$ |
| for $i \in [n]$ | $T(B_1(t_i)) = \{t_i, a\} \qquad T(B_1(f_i)) = \{f_i, a\}$ |
| | $T(V(G)) = T(B_1(a)) = \{t_i, t_j\}$, for $i, j \in [n]$, $i \neq j$ such that there is no |
| | $k \in [m]$ where both $x_i$ and $x_j$ appear in $c_k$.[a] |

Table 1: Positive teaching map for $\mathcal{B}$.

[a]We can assume the existence of such a pair $i, j$, because introducing an artificial variable and a unique clause in which it occurs gives an equivalent instance. Indeed, the artificial variable appears only once and its assignment can be chosen so that the added clause is satisfied without the rest of the formula being affected.

| | $s_k^*$ | $s_k^{**}$ | $s_k^{***}$ | $s_k'$ | $s_k''$ | $s_0'$ | $t_i$ | $f_i$ |
|---|---|---|---|---|---|---|---|---|
| $r_i^*$ | $v$ | $r_0'$ | $r_0'$ | $r_i'$ | $r_i'$ | $r_i'$ | $r_i'$ | $r_i'$ |
| $r_i^{**}$ | $s_0'$ | $r_0'$ | $r_0'$ | $s_0'$ | $s_0'$ | $r_i'$ | $r_i'$ | $r_i'$ |
| $r_i^{***}$ | $s_0'$ | $r_0'$ | $r_0'$ | $s_0'$ | $s_0'$ | $r_i''$ | $r_i''$ | $r_i''$ |
| $r_i'$ | $s_k'$ | $r_0'$ | $r_0'$ | $r_i'$ | $r_i'$ | $r_i'$ | $r_i'$ | $r_i'$ |
| $r_i''$ | $s_k'$ | $r_0'$ | $r_0'$ | $r_i''$ | $r_i''$ | $r_i''$ | $r_i''$ | $r_i''$ |
| $r_0'$ | $s_k'$ | $s_k'$ | $s_k''$ | $s_k'$ | $s_k''$ | $r_0'$ | $r_0'$ | $r_0'$ |
| $t_j$ | $s_k'$ | $s_k'$ | $s_k''$ | $s_k'$ | $s_k''$ | $s_0'$ | $t_i$ | $f_i$ |
| $f_j$ | $s_k'$ | $s_k'$ | $s_k''$ | $s_k'$ | $s_k''$ | $s_0'$ | $t_i$ | $f_i$ |

(a) Here $i, j \in [n]$, $k \in [m]$, and $v \in \{t_i, f_i\} \setminus N(s_k^*)$. According to Table 1, $\{t_i, f_i\} \setminus N(s_k^*) = t_i$ if $x_i = \text{True}$ satisfies $c_k$; and $\{t_i, f_i\} \setminus N(s_k^*) = f_i$ if $x_i = \text{False}$ satisfies $c_k$.

| | $s_k^*$ | $s_k^{**}$ | $s_k^{***}$ | $s_k'$ | $s_k''$ |
|---|---|---|---|---|---|
| $s_l^*$ | $s_k'$ | $v$ | $s_k'$ | $v$ | $v$ |
| $s_l^{**}$ | $s_k'$ | $s_k'$ | $s_k'$ | $r_0'$ | $r_0'$ |
| $s_l^{***}$ | $s_k'$ | $s_k'$ | $s_k''$ | $r_0'$ | $r_0'$ |
| $s_l'$ | $s_k'$ | $s_k'$ | $s_k''$ | $s_k'$ | $s_k'$ |
| $s_l''$ | $s_k'$ | $s_k'$ | $s_k''$ | $s_k'$ | $s_k''$ |
| $s_0'$ | $s_0'$ | $s_0'$ | $s_0'$ | $s_0'$ | $s_0'$ |

(b) Here, $k, l \in [m]$; filled cells correspond to the case $k = l$, and the others to $k \neq l$. Here, $v$ is any vertex in $N(s_l^*) \cap A$. The table for vertices of the variable force-gadgets is defined similarly, interchanging all $s$ and $r$ symbols.

Table 2: For each $u, z \in V(G)$, in a cell at the intersection of the corresponding row and column, we place a vertex $w \in V(G)$ such that $w \in T(B_1(u)) \cup T(B_1(z))$ and $w \notin B_1(u) \cup B_1(z)$ (according to the teaching map defined by Table 1).

is a vertex (either $t_i$ or $t_j$) that is in $T(V(G))$ but not in $B_1(u)$ as there is no clause force-gadget that would be attached to both $t_i$ and $t_j$.

Thus, we showed that for any pair of balls in $\mathcal{B}$, both necessary conditions for the defined teaching sets hold. Hence, the defined positive teaching map is non-clashing for $\mathcal{B}$. $\qquad\square$

**Lemma 3.** *If $(G, 2)$ is a YES-instance of* STRICT NON-CLASH, *then $\phi$ is a YES-instance of 3-*SAT.

*Proof.* Let $T$ be a positive non-clashing teaching map of dimension 2 for the set $\mathcal{B}$ of all balls in $G$. By definition, for each pair of distinct balls $B_1, B_2 \in \mathcal{B}$, there is $w \in T(B_1) \cup T(B_2)$ such that $w \notin B_1 \cap B_2$. For each $i \in [n]$, consider a pair of vertices $r_i^*$ and $r_i^{***}$. By the construction, $B_1(r_i^{***}) \subset B_1(r_i^*)$ and $r_i'$ is the unique vertex in $B_1(r_i^*)$ that is not in $B_1(r_i^{***}) \cap B_1(r_i^*)$. Thus, since $T$ is a positive non-clashing teaching map for $\mathcal{B}$, $r_i'$ is in $T(B_1(r_i^*))$. Now, consider a pair of vertices $r_i^*$ and $r_i^{**}$. Similarly, $B_1(r_i^{**}) \subset B_1(r_i^*)$ and $\{t_i, f_i\} = B_1(r_i^*) \setminus B_1(r_i^{**})$. So, for $T$ to distinguish $B_1(r_i^{**})$ and $B_1(r_i^*)$, exactly one of $t_i$ and $f_i$ (as $T(B_1(r_i^*))$ already contains $r_i'$) is in

$T(B_1(r_i^*))$. The same arguments work for clause force-gadgets, by symmetry of the construction, and we obtain that, for each $k \in [m]$, there is $v \in B_1(s_k^*) \cap A$ such that $T(B_1(s_k^*)) = \{s_k', v\}$.

Now, let us use the fact that for $i \in [n]$ and $k \in [m]$, the balls $B_1(r_i^*)$ and $B_1(s_k^*)$ are distinguished by $T$. If $B_1(s_k^*) \cap \{t_i, f_i\} = \emptyset$, whichever of $t_i, f_i$ that is in $B_1(r_i^*)$ distinguishes the two balls. However, as we have shown before, for each $k \in [m]$, $|T(B_1(s_k^*)) \cap A| = 1$. So, there exists $i \in [n]$ such that $B_1(s_k^*) \cap \{t_i, f_i\} \neq \emptyset$. W.l.o.g., let us assume that $B_1(s_k^*) \cap \{t_i, f_i\} = f_i$ (which means that $T(B_1(s_k^*)) = \{s_k', f_i\}$). As a result, the only valid option for $B_1(r_i^*)$ to be distinguished from $B_1(s_k^*)$ is that $T(B_1(r_i^*)) = \{r_i', t_i\}$. In the other symmetric case where $B_1(s_k^*) \cap \{t_i, f_i\} = t_i$, we obtain $T(B_1(r_i^*)) = \{r_i', f_i\}$.

Let us now define an assignment $\tau : \mathcal{X} \to \{\texttt{True}, \texttt{False}\}$ in the following way. For each $i \in [n]$, if $t_i \in T(B_1(r_i^*))$, we set $\tau(x_i) = \texttt{True}$; otherwise $f_i \in T(B_1(r_i^*))$ and we set $\tau(x_i) = \texttt{False}$. Let us show that $\tau$ indeed satisfies the 3-SAT instance $\phi$. As we discussed above, for each $k \in [m]$, $T(B_1(s_k^*))$ has an intersection with $A$ in exactly one vertex, w.l.o.g., let it again be that $T(B_1(s_k^*)) \cap A = \{f_i\}$. Then, $T(B_1(r_i^*)) \cap A = \{t_i\}$. So, $\tau(x_i) = \texttt{True}$. By our reduction, $s_k^*$ is adjacent to $f_i$ if assigning $\texttt{False}$ to $x_i$ **does not** satisfy $c_k$, while assigning $\texttt{True}$ to $x_i$ would. Thus, the assignment $\tau$ satisfies all the $m$ clauses of the initial 3-SAT instance $\phi$. $\square$

The proof of Theorem 1 then follows from Lemma 2 and Lemma 3. In particular, they prove that there is a polynomial-time reduction which transforms any instance of 3-SAT with $n$ variables and $m$ clauses into an equivalent instance $(G, 2)$ of STRICT NON-CLASH where $|V(G)| = \mathcal{O}(n + m)$ and $G$ is a split graph of diameter 2. The properties of this reduction also allow us to infer more precise algorithmic lower bounds. In particular, since an algorithm solving STRICT NON-CLASH in $2^{o(|V(G)| \cdot d \cdot k)}$ time would allow us to solve 3-SAT in $2^{o(n+m)}$ time:

**Theorem 4.** *Unless the Exponential Time Hypothesis fails, there is no algorithm solving* STRICT NON-CLASH *in time* $2^{o(|V(G)| \cdot d \cdot k)}$, *where $d$ and $k$ are the diameter of $G$ and the target positive non-clashing teaching dimension of the instance, respectively.*

We complement this lower bound with a refined upper bound for the more general NON-CLASH:

**Proposition 5.** NON-CLASH *can be solved in* $2^{\mathcal{O}(|V(G)| \cdot d \cdot k \cdot \log |V(G)|)}$ *time.*

*Proof.* We can assume that $G$ is connected, as otherwise we can solve NON-CLASH independently on each of the connected components of $G$. For any $v \in V(G)$ and $r \in \mathbb{N}$, there are at most $\binom{|V(G)|}{k} = \mathcal{O}(2^{k \cdot \log |V(G)|})$ possible choices for $T(B_r(v))$, and there are at most $\mathcal{O}(|V(G)| \cdot d)$ unique balls in $G$. Due to the latter, for each possible teaching map, we can check in polynomial time whether it is a positive teaching map that satisfies the non-clashing teaching property. Thus, there is a brute-force algorithm running in $2^{\mathcal{O}(|V(G)| \cdot d \cdot k \cdot \log |V(G)|)}$ time. $\square$

## 4 FIXED-PARAMETER TRACTABILITY VIA VERTEX INTEGRITY

Given that NON-CLASH is NP-hard, it is natural to ask whether the problem can be solved efficiently on inputs exhibiting some well-defined structural properties. In this section, we establish the fixed-parameter tractability of NON-CLASH when parameterized by the vertex integrity of the input graph. Consider an instance $(G, \mathcal{B}, k)$ of NON-CLASH and let $p$ be the vertex integrity of $G$. As the first step, we invoke the known algorithm to compute a "witness" for the vertex integrity in time $p^{\mathcal{O}(p)} |V(G)|$ (Fellows & Stueckle, 1989), *i.e.*, a set $X \subset V(G)$ such that $|V(H) \cup X| \leq p$ for each connected component $H$ of $G - X$. Let $\mathcal{H}$ denote the set of connected components of $G - X$. To make use of the vertex integrity of $G$, we will partition the components of $\mathcal{H}$ into a parameter-bounded number of equivalence classes such that the elements belonging to the same class share some structural properties that will allow us to consider them, to some extent, interchangeable.

**Definition 1.** *Two subgraphs $H, H' \in \mathcal{H}$ are* twin-blocks *with respect to $\mathcal{B}$, denoted $H \sim_{\mathcal{B}} H'$, if there exists an isomorphism $\alpha_{H,H'}$ from $H$ to $H'$ with the following properties:*

- *for each $u \in V(H)$ and $v \in X$, $uv \in E(G)$ if and only if $\alpha_{H,H'}(u)v \in E(G)$, and*

- *for each $u \in V(H)$ and $r \in \mathbb{N}$, $B_r(u) \in \mathcal{B}$ if and only if $B_r(\alpha_{H,H'}(u)) \in \mathcal{B}$.*

Intuitively, $H \sim_{\mathcal{B}} H'$ if and only if there is a bijection $\alpha_{H,H'}$ between the vertices of the two subgraphs which preserves (1) edges inside $H$ and $H'$, (2) edges to $X$, and (3) the existence of balls in $\mathcal{B}$ centered at the vertices of $H$ and $H'$. We refer to $\alpha_{H,H'}$ as the *canonical isomorphism* between the two twin-blocks at hand, and if multiple choices of $\alpha$ exist, we choose and fix one arbitrarily; we drop the indices of $\alpha$ when the subgraphs are clear from the context. Observe that for any choice of $H$ and $H'$, $H \sim_{\mathcal{B}} H'$ can be tested in time at most $p^{\mathcal{O}(p)}$ by enumerating all possible choices of $\alpha$.

Clearly, $\sim_{\mathcal{B}}$ is an equivalence relation and we denote by $[H]_{\sim_{\mathcal{B}}}$ the equivalence class containing $H$. For $u \in V(H)$, we further define $[u]_{\sim_{\mathcal{B}}} = \{\alpha_{H,H'}(u) \mid H' \in [H]_{\sim_{\mathcal{B}}}\}$, and similarly for $B_r(u) \in \mathcal{B}$, $[B_r(u)]_{\sim_{\mathcal{B}}} = \{B_r(u') \in \mathcal{B} \mid u' \in [u]_{\sim_{\mathcal{B}}}\}$; intuitively, these refer to the sets of counterparts of $u$ and $B_r(u)$ in the equivalence class, respectively. For brevity, we overload the notation $\sim_{\mathcal{B}}$ and use $v \sim_{\mathcal{B}} w$ (or $B_r(v) \sim_{\mathcal{B}} B_r(w)$) to express that $v \in [w]_{\sim_{\mathcal{B}}}$ (or $B_r(v) \in [B_r(w)]_{\sim_{\mathcal{B}}}$, respectively).

**Observation 6.** *The number of equivalence classes on $\mathcal{H}$ defined by $\sim_{\mathcal{B}}$ is at most $2^{\mathcal{O}(p^3)}$.*

*Proof.* All graphs in $\mathcal{H}$ have size at most $p$, which means that the number of non-isomorphic graphs in $\mathcal{H}$ can be trivially upper-bounded by $p \cdot 2^{p^2}$. Since $|X| < p$, there are also at most $p^2$ possible edges between $X$ and any $H \in \mathcal{H}$ in $G$. Lastly, the number of balls centered in $H \in \mathcal{H}$ is bounded above by the number of vertices in $H$ times the diameter of $G$, which is $p \cdot \mathcal{O}(p^2) = \mathcal{O}(p^3)$. Indeed, the diameter of a connected graph with vertex integrity $p$ is at most $\mathcal{O}(p^2)$, since the parameter does not increase by taking induced subgraphs and the vertex integrity of a path of length $j$ is $\mathcal{O}(\sqrt{j})$. Combining these elements, we can upper-bound the total number of equivalence classes by $p \cdot 2^{p^2} \cdot 2^{p^2} \cdot 2^{\mathcal{O}(p^3)} = 2^{\mathcal{O}(p^3)}$. $\qquad\qquad\square$

While the equivalence relation $\sim_{\mathcal{B}}$ is defined based on the input (in particular, $G$ and $\mathcal{B}$), our proof requires also considering a more refined equivalence relation based on the structure of a hypothetical positive non-clashing teaching map. Toward this, we use the following notion to capture how a hypothetical teaching set interacts with the balls centered in the components of $\mathcal{H}$.

**Definition 2.** *The* blueprint *$S$ of a teaching set $T(B)$ for a ball $B = B_r(u)$ centered in $H \in \mathcal{H}$ is a tuple $(S_X, S_H, S_f)$ composed of:*

1. *the set $S_X = T(B) \cap X$,*

2. *the set $S_H = T(B) \cap V(H)$,*

3. *the set $S_f = \{f_{H_0} \mid H_0 \in \mathcal{H}\}$ of functions, where for each $H_0$ the function $f_{H_0} : V(H_0) \to \{0, 1, 2\}$ specifies whether for a vertex $v \in V(H_0)$, the set $([v]_{\sim_{\mathcal{B}}} \cap T(B)) \setminus V(H)$ of counterparts of $v$ outside of $H$ has size $0$, $1$ or at least $2$.*

Intuitively, the blueprint specifies how the teaching set for $B$ interacts with (1) the set $X$ and (2) the vertices inside $H$ itself; for the rest of the graph, the blueprint also counts how many "equivalent" vertices it contains from each equivalence class of $\mathcal{H}$, *but only up to* $2$. At this point, it may not be clear why we do not differentiate between any size greater than $2$; the reason is that if the actual size is $3$ or more, there are superfluous elements in the teaching set, as we prove below. In fact, we prove a more general statement which holds regardless of whether vertices in $H$ are counted or not.

**Lemma 7.** *Let $u \in V(G)$, $B = B_r(u) \in \mathcal{B}$, and $T$ be a positive non-clashing teaching map for $\mathcal{B}$. Suppose there exist $H_0 \in \mathcal{H}$ and $v \in V(H_0)$ such that $|[v]_{\sim_{\mathcal{B}}} \cap T(B)| \geq 3$. Then, there exists $z$ in $[v]_{\sim_{\mathcal{B}}} \cap T(B)$ such that removing $z$ from $T(B)$ yields a positive non-clashing teaching map for $\mathcal{B}$.*

*Proof.* Let $v$ and $H_0$ satisfy the premise, and let $z_1, z_2, z_3$ be three distinct elements of $\{w \in T(B) \mid w \in V(H'), H' \sim_{\mathcal{B}} H_0, \alpha(w) = v\}$. We denote by $H_1$ ($H_2$, $H_3$, resp.) the component of $\mathcal{H}$ containing $z_1$ ($z_2$, $z_3$, resp.). By definition, these components are disjoint since the vertices are $\sim_{\mathcal{B}}$-equivalent, and thus, $u$ can be in at most one of $H_1, H_2, H_3$ (and possibly none, *i.e.*, $u$ could be in some other component or in $X$). Without loss of generality, we assume that $u$ is not in $V(H_3)$, and claim that removing $z_3$ from $T(B)$ results in a positive non-clashing teaching map $T'$ for $\mathcal{B}$.

We prove this claim as follows. Toward a contradiction, suppose there is a ball $B' = B_{r'}(u')$ such that $T'$ does not satisfy the non-clashing condition for $B$ and $B'$. Then, $z_3$ was the only vertex in $T(B) \cup T(B')$ that was not contained in $B \cap B'$, and hence, $z_3 \notin B'$ and $z_1, z_2 \in B'$. Therefore,

$d(u', z_1) < d(u', z_3)$, which implies that $u' \in V(H_1)$ since $z_1 \sim_{\mathcal{B}} z_3$. However, we also have that $d(u', z_2) < d(u', z_3)$, and so, $u' \in V(H_2)$, which is a contradiction since $V(H_1) \cap V(H_2) = \emptyset$. Thus, such a ball $B'$ does not exist, and we can safely remove $z_3$ from $T$, proving the lemma. $\qquad\square$

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

For $A \in \{\mathcal{K}, \mathcal{K}_1, \mathcal{K}_2\}$, note that $A$ is a subset of $\mathcal{H}$, but by slight abuse of notation, we will write $x \in V(A)$ for $x \in V(G)$ to denote $\exists H \in A$ with $x \in V(H)$ when not leading to any ambiguity. It is not difficult to see that the definitions yield the following.

**Observation 10.** *The number of components in $\mathcal{K}_2$ is upper-bounded by $c(p) = 2^{2^{\mathcal{O}(p^3)}}$.*

*Proof.* Any $(B, x)$-core $\mathcal{K}_{(B,x)}$ contains at most $s(p) + 1$ components by definition. Thus, any $B$-core contains at most $\mathcal{O}(s(p)^2)$ components (see Corollary 8). Since two $\sim_{\mathcal{B}}$-equivalent balls from perfectly-equivalent twin-blocks have the same $B$-core, we get that $\mathcal{K}$ has at most $\mathcal{O}(p^3) \cdot 2^{2^{\mathcal{O}(p^3)}} \cdot \mathcal{O}(s(p)^2) = 2^{2^{\mathcal{O}(p^3)}}$ components (cf. Observation 9). Adding $p$ to this value before multiplying it by $\mathcal{O}(p^3)$ gives a bound on the number of balls centered in $\mathcal{K}$ or $X$, and with an additional factor $s(p)$ we get the maximum size of the union of their teaching sets, which is an upper bound for the number of components in $\mathcal{K}_1$. We can reiterate this to obtain an upper bound for $|\mathcal{K}_2|$ which is:

$$s(p) \cdot \mathcal{O}(p^3) \cdot \left(p + s(p) \cdot \mathcal{O}(p^3) \cdot (p + 2^{2^{\mathcal{O}(p^3)}})\right) = 2^{2^{\mathcal{O}(p^3)}}. \qquad \square$$

We can now formalize the notion of "well-behaved" teaching maps via the property of compactness.

**Definition 7.** *A teaching map $T$ is* compact *if, for any $H \in \mathcal{H} \setminus \mathcal{K}_2$ and any $x \in V(H)$, each ball whose teaching set contains $x$ is centered in $H$.*

Our next aim is to prove that we can safely restrict our attention to computing compact teaching maps only; this is stated in Lemma 13 later on. Before we can establish that result, we first prove two auxiliary statements.

**Lemma 11.** *Let $T$ be a positive non-clashing teaching map for $\mathcal{B}$. Let $B \in \mathcal{B}$ be centered in $H_B \in \mathcal{H} \setminus \mathcal{K}_1$ with $x \in T(B) \cap V(H)$ for some $H \in \mathcal{H} \setminus H_B$ and $H_i \in \mathcal{K}_{(B,x)}$. By definition, there exists $x_i \in [x]_{\sim_{\mathcal{B}}} \cap V(H_i)$. If replacing $x$ by $x_i$ in $T(B)$ creates a conflict, then there exists $z_i \in T(B) \cap V(H_i)$.*

*Proof.* Let $B'$ be a ball creating a conflict with $B$ when we replace $x$ by $x_i$ in $T(B)$. For a conflict to happen, it means that $x_i \in B'$ whereas $x \notin B'$. Since $x_i \in [x]_{\sim_{\mathcal{B}}}$, then $B'$ has to be centered in $H_i$. Let us consider $B'_0 \in [B']_{\sim_{\mathcal{B}}}$ centered in $H$. Prior to the replacement, $B'_0$ has no conflict with $B$, and thus, there are two possibilities:

(1) Let us first consider the case where there exists $z \in T(B'_0)$ that distinguishes $B'_0$ from $B$. By definition, $z \notin B$. Since $B' \sim_{\mathcal{B}} B'_0$ and $H' \equiv^T_{\mathcal{B}} H$, there exists $z' \in T(B')$, such that $z' \in [z]_{\sim_{\mathcal{B}}}$. As $z'$ does not distinguish $B'$ from $B$, it holds that $z' \in B$. However, since $z' \sim_{\mathcal{B}} z$, this means that $z' \in H_B$, which leads to a contradiction since $H_B \notin \mathcal{K}_1$.

(2) Otherwise, there exists $z \in T(B)$ that distinguishes $B$ from $B'_0$, which means that $z \notin B'_0$. However, $z \in B'$ since there is a conflict with $B'$. As $B' \sim_{\mathcal{B}} B'_0$ and $H' \equiv^T_{\mathcal{B}} H$, then the facts that $z \notin B'_0$ and $z \in B'$ imply that $z \in H_i$. Hence, we found the $z_i$ whose existence we claimed. $\quad \square$

**Lemma 12.** *Let $T$ be an optimal positive non-clashing teaching map for $\mathcal{B}$, with $B \in \mathcal{B}$ centered in $H_B \in \mathcal{H} \setminus \mathcal{K}_1$. For $x \in T(B) \cap V(H)$ for some $H \in \mathcal{H} \setminus H_B$, if $|\mathcal{K}_{(B,x)}| = s(p) + 1$, then there are $H_i \in \mathcal{K}_{(B,x)}$ and $x_i \in [x]_{\sim_{\mathcal{B}}} \cap V(H_i)$ such that replacing $x$ by $x_i$ in $T(B)$ creates no conflict.*

*Proof.* We prove the lemma by contradiction. Suppose that for all $x_i \in [x]_{\sim_{\mathcal{B}}}$ of $\mathcal{K}_{(B,x)}$, a conflict is created. By Lemma 11, for all $H_i$ in $\mathcal{K}_{(B,x)}$, there exists $z_i \in T(B) \cap V(H_i)$. However, these $H_i$'s are all disjoint, and hence, $|T(B)| \geq |\mathcal{K}_{(B,x)}| = s(p) + 1$, which contradicts the maximum size of a teaching set established in Corollary 8. This proves that there exist such an $H_i$ and $x_i$. $\quad \square$

**Lemma 13.** *If $\mathcal{B}$ has positive non-clashing teaching dimension $k$, then it also admits a compact positive non-clashing teaching map of dimension $k$.*

*Proof.* This directly follows from the repeated application of Lemma 12 to any positive non-clashing teaching map $T$ for $\mathcal{B}$ of dimension $k$. Let us assume that we do not yet have a compact teaching map. By definition, there exists $H_v \in \mathcal{H} \setminus \mathcal{K}_2$, $v \in V(H_v)$, and $B_r(u)$ such that $v \in T(B_r(u))$ and $u \notin V(H_v)$. It cannot be that $B_r(u)$ is centered in $\mathcal{K}_1$, since otherwise $v \in T(B_r(u))$ would imply that $H_v \in \mathcal{K}_2$. We can then apply Lemma 12 to $B_r(u)$ and $v$, and replace $v$ in $T(B_r(u))$ with some vertex in $\mathcal{K}$. This preserves the non-clashing quality of the teaching map, and does not increase the size of the teaching sets. Moreover, since the number of such pairs $v, B_r(u)$ strictly decreases with each application, we eventually reach the point where there is no such occurrence in the teaching map, meaning that the obtained positive non-clashing teaching map is compact. $\square$

We now proceed to the crux of our algorithm: the proof that one can reduce the size of the instance $(G, \mathcal{B})$ without changing its positive non-clashing teaching dimension (formalized in Lemma 15). Toward this, it will be useful to focus on how the balls in $\mathcal{B}$ interact with certain subgraphs of $G$.

**Definition 8.** *Let $G'$ be an induced subgraph of $G$. Then, $\mathcal{B}$ induces the set $\mathcal{B}'$ of vertex sets w.r.t. $G'$, where $\mathcal{B}' = \{B_r(u) \cap V(G') \mid B_r(u) \in \mathcal{B} \wedge u \in V(G')\}$.*

We note that $\mathcal{B}'$ need not necessarily be a set of balls in $G'$ itself: for instance, it may well happen that for some $B \in \mathcal{B}'$ the vertex set $B \cap V(G')$ is not even connected. However, under some conditions on $G'$ that we will be able to guarantee, $\mathcal{B}'$ is, in fact, a set of balls in $G'$.

**Lemma 14.** *Let $G$ be a graph, $\mathcal{B}$ a set of balls in $G$, and $G'$ an induced subgraph of $G$. If $V(G')$ contains $X$ and, for every $H \in \mathcal{H}$, either (1) $V(H) \subseteq V(G')$ or (2) $V(H) \cap V(G') = \emptyset \wedge \exists H', H'' \in [H]_{\sim_\mathcal{B}}, H' \neq H'', V(H') \cup V(H'') \subseteq V(G')$ holds, then the set $\mathcal{B}'$ induced by $\mathcal{B}$ w.r.t. $G'$ is a set of balls in $G'$. Moreover, there is a bijection between balls in $\mathcal{B}'$ and balls in $\mathcal{B}$ centered in $G'$.*

*Proof.* The key element of this proof is that in such a graph $G'$, the distance $d_{G'}(u, v)$ between two vertices $u$ and $v$ is the same as the distance $d_G(u, v)$ between them in $G$. Indeed, for $u, v \in V(G')$, suppose that the shortest path between them in $G$ uses some vertices not present in $G'$. For a consecutive set of such missing vertices on the path, since they are missing in $G'$ and connected in $G$, they are not in $X$ and they are all in the same component $H \in \mathcal{H}$. Since $H$ is missing in $G'$, the vertices directly before and after on the path are from $X$: these vertices exist because $u, v \in V(G')$. Since there is $H' \in [H]_{\sim_\mathcal{B}}$ such that $V(H') \subseteq V(G')$, we will replace the vertices of the path in $H$ by the $\sim_\mathcal{B}$-equivalent vertices of $H'$. The newly constructed path has the same length, is still a path by the definition of $H \sim_\mathcal{B} H'$, and has strictly fewer vertices not in $G'$. Iterating this argument until we obtain a path with only vertices of $G'$, we prove that the distance between $u$ and $v$ is the same in $G'$ as it was in $G$. From now on, we denote this distance $d(u, v)$, without referring to the graph.

We are now ready to prove that the vertex sets in $\mathcal{B}'$ are indeed balls in $G'$. Let $B' \in \mathcal{B}'$. There exist $u \in V(G')$ and $B_r(u) \in \mathcal{B}$, such that $B' = B_r(u) \cap V(G') = \{v \in V(G'), d(u, v) \leq r\}$. Hence, $B'$ is indeed a ball in $G'$. It remains to prove that this construction is injective. Let $u, v \in V(G')$ and $B_r(u), B_{r'}(v) \in \mathcal{B}$, such that $B_r(u) \cap V(G') = B_{r'}(v) \cap V(G')$ and assume there is $w \in V(G) \setminus V(G')$ such that $w \in B_r(u)$ and $w \notin B_{r'}(v)$. Let $H \in \mathcal{H}$ be such that $w \in V(H)$, and $H', H'' \in [H]_{\sim_\mathcal{B}}$ are the two corresponding components whose existence is required by (2), and $w' \in V(H') \cap [w]_{\sim_\mathcal{B}}$ ($w'' \in V(H'') \cap [w]_{\sim_\mathcal{B}}$, resp.). If $w' \notin B_r(u)$, then it implies that $d(u, w) < d(u, w')$, and thus, $u \in V(H)$, which is a contradiction since $V(G') \cap V(H) = \emptyset$. With the same argument for $w''$, we infer that $w', w'' \in B_r(u)$, and hence, $w', w'' \in B_{r'}(v)$ since $w', w'' \in V(G')$. However, $w' \in B_{r'}(v)$ implies that $d(v, w') < d(v, w)$, which in turn implies that $v \in V(H')$. We also obtain $v \in V(H'')$ by the same reasoning with $w''$, and this leads to a contradiction since $V(H') \cap V(H'') = \emptyset$. We proved that it was impossible to find two such balls, which proves the claimed bijection between balls in $\mathcal{B}'$ and balls in $\mathcal{B}$ centered in $G'$. $\square$

We define the *reduced graph $G'$ of $G$* as the graph obtained from $G$ by removing all but $f(p) = c(p) + \binom{p \cdot c(p) + p}{s(p)}^{b(p)} + 1$ twin-blocks from each large class of $\sim_\mathcal{B}$, where $b(p) = \mathcal{O}(p^3)$ is the maximum number of distinct balls centered in any component of $\mathcal{H}$. Let $\mathcal{B}'$ be the set induced by $\mathcal{B}$ on $G'$ according to Definition 8. By Lemma 14, $\mathcal{B}'$ is a set of balls in $G'$. The size of $G'$ is at most $g(p) = p + f(p) \cdot 2^{\mathcal{O}(p^3)} = 2^{2^{\mathcal{O}(p^3)}}$. Thus, an optimal positive non-clashing teaching map for $\mathcal{B}'$ can be computed (*e.g.*, by brute force) in time that depends only on $p$. We now aim at proving that $\mathcal{B}'$ is equivalent to $\mathcal{B}$, *i.e.*, that the optimal positive non-clashing teaching dimension is the same for both.

**Lemma 15.** *Suppose that there is a solution for $(G, \mathcal{B}, k)$. Then, there exists a solution $T'$ for the reduced instance $(G', \mathcal{B}', k)$.*

*Proof.* Applying Lemma 13 gives us the existence of a compact solution $T$. Up to renaming, $G'$ corresponds to $G$ in which we remove only components **not** in $\mathcal{K}_2$ for $T$. Let us define $T'$ as the restriction of $T$ to $\mathcal{B}'$—this is well-defined since each ball in $\mathcal{B}'$ corresponds to exactly one in $\mathcal{B}$ as per Lemma 14. It is easy to see that if two balls in $\mathcal{B}'$ were in conflict, then the corresponding balls in $\mathcal{B}$ would be as well. Indeed, the teaching sets are exactly the same, since we did not remove any component of $\mathcal{K}_2$ and $T$ is compact. Furthermore, the balls themselves can only be smaller in $\mathcal{B}'$, and thus, the element used by $T$ to distinguish between the two balls of $\mathcal{B}$ can be used by $T'$ to distinguish between the two balls in $\mathcal{B}'$. $\qquad\square$

The following lemma establishes that a solution for the reduced instance can be lifted to one for the original instance; note that this also ensures that our algorithm will be constructive.

**Lemma 16.** *Let $k \in \mathbb{N}$ and $T'$ be a compact solution for the instance $(G', \mathcal{B}', k)$. We can construct a compact solution $T$ for $(G, \mathcal{B}, k)$ in $2^{2^{\mathcal{O}(p^3)}} \cdot |V(G)|^{\mathcal{O}(1)}$ time.*

*Proof.* We iteratively add back the missing components $(H_0, \ldots, H_x)$ of $\mathcal{H}$ and update the set of balls accordingly. We start by choosing an arbitrary ordering $\prec$ on $\mathcal{H}$ such that: $\forall H, H' \in \mathcal{H}, V(H) \subseteq V(G') \wedge V(H') \cap V(G') = \emptyset \Rightarrow H \prec H'$. We then compute $\mathcal{K}_2^{T'}$ for $T'$ and $\prec$ on $G'$, as it is used at each step. The time necessary for it depends only on $p$ and is $\mathcal{O}(|\mathcal{B}'| \cdot s(p))$, as it suffices to check for each ball which components its teaching set uses, apart from the component where it is potentially centered. Indeed, as $T'$ is compact, $H$ being in $\mathcal{K}_2^{T'}$ implies there is a ball not centered in $H$ using $x \in V(H)$ in its teaching set. We prove the lemma via the following claim.

**Claim 17.** *Let $0 \le c < x$, $\mathcal{B}^*$ be the set of balls induced by $\mathcal{B}$ on $G^* = G[V(G') \bigcup_{0 \le i < c} V(H_i)]$, and let $T^*$ be a solution for $(G^*, \mathcal{B}^*, k)$ such that $\mathcal{K}_2^{T'}$ is the 2-extended core for $T^*$. We can construct, in $2^{2^{\mathcal{O}(p^3)}} \cdot |V(G)|^{\mathcal{O}(1)}$ time, a solution $T^+$ for $(G^+, \mathcal{B}^+, k)$, where $G^+ = G[V(G^*) \cup V(H_c)]$, $\mathcal{B}^+$ is induced by $\mathcal{B}$ on $G^+$, and $\mathcal{K}_2^{T'}$ is the 2-extended core for $T^+$.*

*Proof.* It is easy to observe that $\mathcal{B}^*$ is induced by $\mathcal{B}^+$ on $G^*$. Thus, by Lemma 14, there is a bijection between the balls in $\mathcal{B}^*$ and those of $\mathcal{B}^+$ which are centered in $G^*$. We can compute $\mathcal{B}^+$ in $|V(G)|^{\mathcal{O}(1)} \cdot p^2$ time, since for the at most $|V(G)| \cdot p^2$ balls in $\mathcal{B}^+$, a breadth-first search on $G^+$ suffices to compute it. Note that for any ball $B^* \in \mathcal{B}^*$, the corresponding ball $B^+ \in \mathcal{B}^+$ is a superset of it, and we set $T^+(B^+) = T^*(B^*)$. Note that for any two balls whose teaching set we define in this way, there can be no conflict between them. Indeed, any element previously distinguishing their respective equivalent balls in $\mathcal{B}'$ is in $V(G')$, and thus, will still be at the same distance from each center, meaning it distinguishes the two balls in $\mathcal{B}^+$ as well.

However, not all balls in $\mathcal{B}^+$ are centered in $G^*$: we now need to define $T^+$ for the balls centered in $H_c$. Toward this end, we first need to identify some components in $G^*$ which will be useful to define the teaching map so that it is non-clashing.

We know that there are at least $f(p)$ components of $[H_c]_{\sim_{\mathcal{B}}}$ in $G'$ by definition, and thus, in $G^*$. Since $f(p) = c(p) + \left(\frac{p \cdot c(p)+p}{s(p)}\right)^{b(p)} + 1$, there are at least $\left(\frac{p \cdot c(p)+p}{s(p)}\right)^{b(p)} + 1$ components outside of $\mathcal{K}_2^{T'}$. Since all of these components are twin-blocks, they have the same balls (at most $b(p)$ of them), and by the pigeonhole principle, since each ball has a teaching set consisting of at most $s(p)$ vertices of the components of $\mathcal{K}_2^{T'} \cup X$ and the component the ball is centered in (which contain at most $p \cdot c(p) + p$ vertices combined), at least 2 of them have, for each pair of $\sim_{\mathcal{B}}$-equivalent balls, the exact same teaching sets in $\mathcal{K}_2^{T'} \cup X$ as well as isomorphically identical teaching sets in their own components. Let us denote these two components as $H', H''$.

Let $u \in V(H_c)$, $r \in \mathbb{N}$ be such that $B_r^+(u) \in \mathcal{B}^+$. We "copy" the teaching set $T^*(B_r^*(v))$, where $v \in V(H') \cap [u]_{\sim_{\mathcal{B}}}$. Formally, for $\alpha$ the canonical isomorphism from $H'$ to $H_c$, $T^+(B_r^+(u)) = \{\alpha(w) \mid w \in T^*(B_r^*(v)) \cap H'\} \cup \{w \mid w \in T^*(B_r^*(v)), w \notin H'\}$. Thus, from the point of view of any ball centered outside of $H'$ and $H_c$, $B_r^+(u)$ and $T^+(B_r^+(u))$ behave the same as $B_r^*(v)$ and $T^*(B_r^*(v))$, and so, there are no conflicts with such balls, as they were not in conflict with $B_r^*(v)$.

Two balls centered in $H_c$ are also distinguished from each other since this is the case for pairs of balls centered in $H'$. The last thing to check is that balls centered in $H_c$ are distinguished from balls centered in $H'$. Here, we use the fact that balls in $H''$ have the same teaching sets (up to isomorphism between vertices of $H'$ and $H''$) as those in $H'$, and so, if there was a conflict between balls centered in $H'$ and $H_c$, there would also be one between balls centered in $H'$ and $H''$.

Thus, we managed to construct a teaching map $T^+$ for $\mathcal{B}^+$. Since the new teaching sets are imitations of pre-existing teaching sets in $T^*$, the size constraint is satisfied, and it is easy to check that the solution is still compact. Indeed, the teaching sets of balls centered in $H'$ contained only vertices of $H'$ and $\mathcal{K}_2^{T'}$, and thus, those of balls centered in $H_c$ contain by construction only vertices of $H_c$ and $\mathcal{K}_2^{T'}$. Furthermore, $H_c$ is not in $\mathcal{K}_2$ for $T^+$: indeed, $H_c \notin \mathcal{K}$ because there are already enough components $\sim_{\mathcal{B}}$ equivalent to $H_c$ in $G'$, and all of them are preceding $H_c$ in the order $\prec$. Moreover, no vertex of $H_c$ has been added to the teaching sets of the balls not centered in $H_c$: by definition, $H_c \notin \mathcal{K}_2$. Thus, $\mathcal{K}_2 = \mathcal{K}_2^{T'}$.

Note that we can find $H'$ and $H''$ in time $f(p)^2 \cdot (c(p) + p)^{s(p) \cdot \mathcal{O}(p^3)}$, and then we construct teaching sets for the at most $b(p)$ balls in $H_c$, each of them being a simple copy of the equivalent teaching set in $H'$, which can be done in $\mathcal{O}(s(p))$ time. Thus, the running time is $f(p)^2 \cdot (c(p) + p)^{s(p) \cdot \mathcal{O}(p^3)} \mathcal{O}(s(p)) \cdot |V(G)|^{\mathcal{O}(1)} = 2^{2^{\mathcal{O}(p^3)}} \cdot |V(G)|^{\mathcal{O}(1)}$ and we have proven the claim. $\qquad\square$

We can now finish proving the lemma by induction using Claim 17. Since the hypothesis trivially holds for $c = 0$ (by considering $G^* = G'$), we prove the claimed result, and the total running time follows from the fact that the number of components of $\mathcal{H}$ missing in $G'$ are at most $|V(G)|$. $\qquad\square$

Finally, we obtain our main parameterized tractability result by combining the previous ingredients. In particular, in the proof, we construct $(G', \mathcal{B}', k)$, solve the problem there, and argue correctness.

**Theorem 18.** NON-CLASH *is* FPT *parameterized by the vertex integrity of the input graph $G$.*

*Proof.* Let $G$ be a graph with vertex integrity $p$, $\mathcal{B}$ a set of balls in $G$, and $k \in \mathbb{N}$. We provide an algorithm computing a positive non-clashing teaching map for $\mathcal{B}$ of dimension at most $k$, or correctly outputing that none exists, in $q(p) \cdot |V(G)|^{\mathcal{O}(1)}$ time, where $q$ is an elementary function. It is useful to note that if $k \geq s(p)$, then Corollary 8 implies there is always a solution. Moreover, replacing the value of $k$ by $s(p)$ for the rest of the algorithm does not hurt: the solution still exists and has dimension smaller than $k$. Hence, we assume in the rest of the proof that $k \leq s(p)$.

The first step of the actual algorithm is to compute the witness $X \subset V(G)$ for the vertex integrity and the corresponding set $\mathcal{H}$ of connected components. Next, we classify the elements of $\mathcal{H}$ w.r.t. the equivalence classes defined by $\sim_{\mathcal{B}}$ (see Definition 1). Since there are at most $2^{\mathcal{O}(p^3)}$ equivalence classes and the equivalence between two components can be tested in $p^{\mathcal{O}(p)}$ time, we can compute the equivalence classes in $|V(G)| \cdot 2^{\mathcal{O}(p^3)} \cdot p^{\mathcal{O}(p)}$ time with brute force.

We are now ready to compute the reduced graph $G'$ of $G$. We recall that to this end it suffices to remove some arbitrary components of $\mathcal{H}$ whose equivalence class are bigger than $f(p)$, which can be done in linear time. Recalling Definition 8, the set $\mathcal{B}'$ induced by $\mathcal{B}$ on $G'$ can be computed in $|V(G')| \cdot \mathcal{O}(p^2) \cdot |V(G)| \cdot |V(G')|$ time; indeed, it suffices to check for $u \in V(G')$ and $r \in \mathbb{N}$ (which is at most the diameter of the graph) whether $B_r(u)$ exists in $\mathcal{B}$, and then to compute the intersection of it with $V(G')$. The size of the instance $(G', \mathcal{B}', k)$ is upper-bounded by a function of $p$, meaning that we can compute a positive non-clashing teaching map of dimension at most $k$ for $\mathcal{B}'$ in time depending only on $p$ (or determine that none exists)—see Proposition 5.

Using Lemma 15, we know that if there is no such teaching map for $\mathcal{B}'$, there is also none of dimension $k$ for $\mathcal{B}$: the algorithm can safely output that no solution exists. Conversely, if we obtain a positive non-clashing teaching map for $\mathcal{B}'$, then we can use Lemma 16 to construct a positive non-clashing teaching map of dimension at most $k$ for $\mathcal{B}$

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

. We construct a positive teaching map $T$ as follows. In the NAE-satisfying variable assignment for $\phi$, for each variable $x \in \mathcal{X}$, if the integer $j$ is assigned to $x$, then place $x_j$ in $T(V(G))$. For each $c \in \mathcal{C}$, set $T(B_{r_c}(c)) := \{c\}$ and $T(B_{r'_{c'}}(c')) := \{c'\}$. Lastly, for each $x \in X$, $T(B_{6d}(f_x))$ contains $f_x$ and one arbitrary vertex from each of the $k - 1$ variable paths it contains. We prove that $T$ is non-clashing for $\mathcal{B}$.

For any two balls $B_1, B_2 \in \mathcal{B}$ centered at vertices of the form $c$ or $c'$ for the same or different clauses, $T$ satisfies the non-clashing condition since $T(B_1)$ contains its center (some $c$ or $c'$) while $B_2$ does not contain this vertex. Indeed, the radius of $B_2$ is less than $5d$, while the distance between any two vertices of the form $c$ or $c'$ is at least $6d$ since any shortest path between them contains a vertex from a variable path. For any $x, y \in \mathcal{X}$, $T$ satisfies the non-clashing condition for $B_{6d}(f_x)$ and $B_{6d}(f_y)$ since $T(B_{6d}(f_x))$ contains a vertex in $P^y$ while $B_{6d}(f_y)$ does not. For any $x \in \mathcal{X}$ and $c \in \mathcal{C}$, $T$ satisfies the non-clashing condition for $B_{6d}(f_x)$ and $B_{r_c}(c)$, as well as $B_{6d}(f_x)$ and $B_{r'_{c'}}(c')$, since $T(B_{6d}(f_x))$ contains $f_x$ while $B_{r_c}(c)$ and $B_{r'_{c'}}(c')$ do not as $r_c, r_{c'} < 5d$ while $c$ and $c'$ are at distance $6d$ from $f_x$. For any $x \in \mathcal{X}$, $T$ satisfies the non-clashing condition for $V(G)$ and $B_{6d}(f_x)$ since $T(V(G))$ contains a vertex in $P^x$ while $B_{6d}(f_x)$ does not. Finally, for any $c \in \mathcal{C}$, $T$ satisfies the non-clashing condition for $V(G)$ and $B_{r_c}(c)$, as well as $V(G)$ and $B_{r_{c'}}(c')$, since, among the variables contained in the clause $c$, $T(V(G))$ contains at least one vertex from one of those variable paths whose index satisfies $c$, and at least one vertex from one of those variable paths whose index does not satisfy $c$. As can be recalled from the construction, this implies that $B_{r_c}(c)$ and $B_{r_{c'}}(c')$ do not contain the respective vertices. Thus, $T$ satisfies the non-clashing property for all pairs of balls in $\mathcal{B}$.

Now, we prove the reverse direction, so suppose that $(G, \mathcal{B}, k)$ is a YES-instance of NON-CLASH and that $T$ is the corresponding teaching map. For all $x \in \mathcal{X}$, in order for $T$ to satisfy the non-clashing condition for $B_{6d}(f_x)$ and $V(G)$, we have that $T(V(G))$ contains at least one vertex from $P^x$ as $B_{6d}(f_x) \subset V(G)$ and $V(G) \setminus B_{6d}(f_x)$ is restricted to the vertices in $P^x$. Since $k = |\mathcal{X}|$, we in fact have that $T(V(G))$ contains exactly one vertex from $P^x$ for all $x \in \mathcal{X}$. Extract a variable assignment for $\phi$ from $T(V(G))$ as follows. For each $x \in X$, assign the variable $x$ the value of the index of the unique vertex contained in both $P^x$ and $T(V(G))$. We prove that this is an NAE-satisfying variable assignment for $\phi$.

W.l.o.g., let $c \in \mathcal{C}$ be a clause containing the variables $x, y, z \in \mathcal{X}$. In order for $T$ to satisfy the non-clashing condition for $V(G)$ and $B_{r_c}(c)$, $T(V(G))$ must contain at least one vertex in $P^x$, $P^y$ or $P^z$ that is not contained in $B_{r_c}(c)$. Analogously, in order for $T$ to satisfy the non-clashing condition for $V(G)$ and $B_{r'_{c'}}(c')$, $T(V(G))$ must contain at least one vertex in $P^x$, $P^y$ or $P^z$ that is not contained

in $B_{r'_{c'}}(c')$. Recall that all of the vertices in $P^x$, $P^y$, and $P^z$ that are not contained in $B_{r_c}(c)$ have respective indices that correspond to the respective variable values that satisfy the clause $c$. Similarly, recall that all of the vertices in $P^x$, $P^y$, and $P^z$ that are not contained in $B_{r_{c'}}(c')$ have respective indices that correspond to the respective variable values that do not satisfy the clause $c$. As these arguments hold for any clause $c \in \mathcal{C}$, the variable assignment extracted above corresponds to an NAE-satisfying variable assignment for $\phi$. $\square$

Now, we are ready to proceed with the proof of Theorem 19.

*Proof of Theorem 19.* Lemma 20 establishes the correctness of the polynomial-time reduction from the beginning of Section 5. To complete the proof, it remains to show that $\mathtt{fvs}(G) + \mathtt{pw}(G) + k$ is bounded above by a function of $|\mathcal{X}|$. This clearly holds for $k$ which is, by definition, $|\mathcal{X}|$. Deleting from $G$ the vertices $x_1$, $x_d$, and $f_x$ for all $x \in \mathcal{X}$ results in an acyclic graph $G'$; in particular $G$ has a feedback vertex set of size $3|\mathcal{X}|$. To establish a bound on $\mathtt{pw}(G)$, it now suffices to show that $G'$ also has bounded pathwidth. Note that $G'$ consists of a set of connected components, each of which is either a subdivided caterpillar (this is what remains of each component containing a variable path) or a vertex (of the form $c$ or $c'$) with multiple pendent subdivided caterpillars and (simple) paths. Since deleting one further vertex from each connected component may only reduce the pathwidth by 1 and we need a single such deletion operation to reach a graph class of constant pathwidth (see Section 2), we also obtain that $\mathtt{pw}(G)$ is bounded by a function of $|\mathcal{X}|$. $\square$

## 6 Concluding Remarks

Our computational upper and lower bounds provide a near-comprehensive understanding of the complexity of computing the positive non-clashing teaching dimension. Apart from our contributions to the previously studied strict setting, we consider it notable that our work is the first to also tackle the complexity of non-clashing teaching in the non-strict setting—*i.e.*, the more general (and arguably more natural) case where not all possible concepts are present.

One open question highlighted by our work concerns the tiny remaining gap between the algorithmic lower and upper bounds obtained in Theorem 4 and Proposition 5. In particular, is there a way to improve the running time of the latter algorithm to $2^{\mathcal{O}(|V(G)| \cdot d \cdot k)}$ and make the bounds tight? More general directions for future work are to perform a similar complexity analysis in the non-positive setting and to consider approximation algorithms.

## Acknowledgements

This work was funded by the Austrian Science Fund (FWF) [10.55776/Y1329 and 10.55776/COE12], the WWTF Vienna Science and Technology Fund (Project 10.47379/ICT22029), the ■ European Union's Horizon 2020 research and innovation COFUND programme Log-iCS@TUWien (grant agreement No 101034440), the Spanish Ministry of Economic Affairs and Digital Transformation and the European Union-NextGenerationEU through the project 6G-RIEMANN (TSI-063000-2021-147), and the Smart Networks and Services Joint Undertaking (SNS JU) under the European Union's Horizon Europe and innovation programme under Grant Agreement No. 101139067 (ELASTIC). Views and opinions expressed are however those of the authors only and do not necessarily reflect those of the European Union (EU). Neither the EU nor the granting authority can be held responsible for them.