# OpenReview forum: "The Computational Complexity of Positive Non-Clashing Teaching in Graphs"
_ICLR.cc/2025/Conference — ICLR 2025 Poster_

### Official Review · Reviewer_86oC · 2024-10-27

**Soundness:** 4
**Presentation:** 3
**Contribution:** 2
**Rating:** 6
**Confidence:** 2

**Summary:**

This paper studies and obtains several theoretical results for the non-clashing teaching dimension of a concept class. The positive, non-clashing teaching dimension of a class $\cal C$, somewhat similar to VC dimension, is the minimum integer $k$ such that the following holds. For each concept in the class there is a unique set of examples associated with it of size up to $k$, so that for every two sets $S_1, S_2$ of examples (for different concepts $C_1, C_2$), the symmetric difference between $S_1$ and $S_2$ is non-zero.

This notion can equivalently be formulated using balls in graphs, where each concept corresponds to a ball of some radius around some vertex (note that some concepts can strictly contain others, e.g., a ball of radius 2 and a ball of radius 1 around the same vertex). The paper obtains several results about this graph theoretic concept:

1. An NP-hardness result for a strict variant of the non-clashing dimension. It is shown that even determining whether the dimension is at most 2 is NP-hard, resolving a question of Chalopin et al. (COLT 2024).

2. Tighter upper and lower bounds improving upon the aforementioned Chalopin et al. work. The upper and lower bound from the aforementioned work are $2^{O(n^2 d)}$ and $2^{\Omega(n d)}$ where $d$ is the graph diameter, and the current work obtains upper and lower bound of $2^{O(ndk)}$ where $k$ is the non-clashing dimension.

3. Parametrized results obtaining complexity that depend polynomially on $n$ and exponentially on parameters like the vertex integrity. There are also hardness results for well known parametrized notions such as path-width, and by extension, treewidth and cliquewidth.

From the technical perspective, the proofs are rather intricate (especially the lower bound ones, which use careful reductions). I did not verify the full details but the proofs seem believable.

**Strengths:**

- Novelty: new results on an emerging learning concept, which address open problems from the recent literature.

- Mathematical proofs seemingly of significant depth.

- The paper is generally well written.

**Weaknesses:**

- Somewhat limited scope, this is a graph theoretic concept that has not been of interest outside a theory sub-community (and as such might be less suitable for a wide audience conference like ICLR).

- The studied notion seems somewhat "too pessimistic to be interesting". There is an exponential lower bound on the running time even prior to this work (which was improved by this work). The parameterized complexity upper bounds, although new, depend on parameters that seem to be very large, probably linear in $n$, in "realistic" graphs.

Edit on the second bullet: the authors' response helped me contextualize better what this notion aims to capture. I am increasing my score by one point, as this was my main qualm about the paper.

**Questions:**

What do you think about the practical applicability of this non-clashing teaching notion? What is the next set of results that would allow the community to better appreciate this notion?

---

> ### Author Response · Authors · 2024-11-19
>
> Thank you for your constructive review. Non-clashing teaching is an established teaching model which is known to achieve optimal efficiency in the terms of the number of required examples. Further, non-clashing teaching is closely related to the important notion of sample compression schemes and we have added a discussion of this in the related work subsection in the updated version of the submission.
>
> Beyond that, we believe that non-clashing teaching could also help make progress towards the long-standing sample compression conjecture. Intuitively, non-clashing teaching maps can provide the compression and reconstruction for samples which contain the teaching set, in particular when dealing with proper sample compression schemes ("proper" means that the returned concept C must be in the concept class). Often, this is the core idea needed to design (proper) sample compression schemes for concept classes, and then further ideas are needed to deal with the cases where these teaching sets are not fully included in the sample. Thus, also understanding the relationship between non-clashing teaching and the VC-dimension could help lead to progress on the sample compression conjecture, which motivates investigating the question of (Kirkpatrick et al., 2019 and Fallat et al., 2023) that asks whether the VC-dimension is an upper bound for the non-clashing teaching dimension. (Chalopin et al., 2024) points out that balls in cacti or planar graphs could be good candidates for concept classes that negatively answer this question, which supports studying balls in graphs as concept classes, a point that we have also added in the related work subsection of the updated version of the submission.
>
> Due to the above, the intuition is that positive non-clashing teaching could be useful in both labeled and unlabeled (proper) sample compression schemes. The next step would be to do a similar analysis for the general non-clashing teaching in which negative examples can be used; which will be useful for labeled (proper) sample compression schemes. As pointed out by another reviewer, approximation algorithms would also be of interest. Thus, in the updated version of the submission, we have added this as a future direction in the conclusion, and also added an inapproximability result that can be inferred from the literature for the non-strict case in the related work subsection, as well as an inapproximability result following from Theorem 1 for the strict case (stated just before Theorem 1).

---

> > ### Comment · Reviewer_86oC · 2024-11-20
> > **Thank you**
> >
> > Thanks to the authors. The response clarified some concerns I had about the usefulness of the notions and I am increasing my score by one point. Please make sure to write more extensively about the general context also in the paper itself.

---

### Official Review · Reviewer_QH2t · 2024-11-02

**Soundness:** 4
**Presentation:** 3
**Contribution:** 3
**Rating:** 8
**Confidence:** 3

**Summary:**

This paper investigates the computational complexity of determining the positive non-clashing teaching dimension for a set of concepts, focusing specifically on its representation as balls within a graph G.
- The authors show NP-hardness for what is known to be the easier subcase of this problem, where all balls are considered (strict non clash).
- They supplement this hardness result with a positive result for the more general case (non clash), showing fixed parameter tractability with respect to the vertex integrity of a graph, strengthening previous work which had shown fixed-parameter tractability with respect to the vertex cover number, which upper bounds the vertex integrity.
- They complement the fixed-parameter tractability result with a hardness result that shows hardness of non-clash with respect to a number of natural alternative parameters: feedback vertex number, pathwidth, and teaching dimension, thus identifying vertex integrity as a key parameter for tractability.

**Strengths:**

- The paper is clearly written, and answers a number of open questions in the area of machine teaching by strengthening existing results on both the hardness of strict-non-clash, and the fixed-parameter tractability/hardness of non-clash.
- While I am not very familiar with this area, these strengthenings seemed significant in that they show hardness for the easiest subcase of the problem, thus providing a very strong and previously unknown hardness result, and additionally introduce a novel parameter that seems to capture the tractability of the problem given that a combination of other related parameters still render it intractable.
- The proof techniques needed to solve these problems are highly non-trivial, relying on a delicate reduction to 3-sat in the case of the NP-hardness result, and a complicated equivalence class argument in the case of the fixed-parameter tractability result.
- I carefully checked the correctness of the proof of Theorem 1 and was satisfied, but due to the length, have to defer to other reviewers for comments about the correctness of the other proofs.

**Weaknesses:**

- As someone unfamiliar with the area of machine teaching, I felt that some of the introduction/preliminaries relied too heavily on assuming familiarity with previous work, and would have benefitted from additional explanation within the paper to make it more standalone. In particular, I would have liked a bit more discussion on the equivalence between arbitrary concept classes and balls in graphs. I wasn't even sure what arbitrary concept class meant in this case (binary? finite domain?), and how the distinction between strict-non-clash vs non-clash would map to concept classes.
- While I was convinced by the novelty/significance of the results, I would defer to someone more knowledgeable about this area on this point, as well as more generally the significance of the question of positive-non-clashing teaching dimension.
- I feel like the related work could have been more substantial, perhaps discussing related problems in other areas of theory/machine learning.
- Minor note: There is a typo in the first sentence of section 5

**Questions:**

- How do these questions of teaching dimension relate to questions studied in one-way communication complexity or sample compression?
- How does allowing for non-positive examples in the teaching set change the hardness of the problem?

---

> ### Author Response · Authors · 2024-11-19
>
> Thank you for your very detailed and positive review. Taken into account your valuable feedback, the updated version of the submission now has the following changes:
>
> - We describe how to represent finite binary concept classes by balls in a graph at the beginning of page 2;
>
> - We make it clear that we are talking about finite binary concept classes;
>
> - We have added to the related work by mentioning inapproximability, further motivation for studying balls in graphs as concept classes, and the relationship between non-clashing teaching and sample compression schemes;
>
> - We corrected the typo in Section 5.
>
> Regarding the distinction between Strict Non-Clash and Non-Clash, when translating a finite binary concept class to a concept class consisting of balls in a graph, one obtains an instance of the non-strict version: not all balls are present (this fact can be seen as further motivation for studying the non-strict version). However, the strict version is/was motivated by it being the most natural concept class associated with the discrete geometric class consisting of balls in a graph, as all the balls are present.
>
> For Question 1: We are not aware of any immediate connections to the questions studied in one-way communication complexity. However, as mentioned above, the updated submission now discusses non-clashing teaching’s relation to sample compression schemes.
>
> For Question 2: This is indeed an interesting direction for future work. As we stated in the related work subsection, it is known that the non-strict version is NP-hard when negative as well as positive examples are present. However, nothing is known with respect to structural graph parameterizations such as the ones from Theorems 17 and 18. The main high-level intuition on why our hardness reductions do not translate to the setting with both positive and negative examples is that it becomes difficult to force certain vertices to lie in certain teaching sets.

---

> > ### Comment · Reviewer_QH2t · 2024-11-21
> >
> > Thanks for clarifying the exposition and answering my questions! The revisions have improved the clarity of the presentation. While the improvements address my earlier concerns effectively, I maintain my original assessment of the paper's contributions and significance. I appreciate the careful attention you've paid to the reviewer feedback and the thoroughness of your revisions.

---

### Official Review · Reviewer_dmLM · 2024-11-04

**Soundness:** 3
**Presentation:** 3
**Contribution:** 3
**Rating:** 8
**Confidence:** 4

**Summary:**

The authors present several complexity results on the computation of the teaching dimension in the model of non-clashing machine Teaching.
Recent results have appeared on the "strict" variant of the problem and the authors consider the more general case of the non-strict variant. For the strict variant they also strengthen the previous hardness result by showing NP-hardness also for deciding teaching dimension constant bounds. Under the ETH this is also extended to upper and lower bounds on an optimal algorithm complexity for the problem.
For the non-strict variant they present new Fixed parameterized complexity results, with respect to less restrictive parameters (Vertex integrity vs vertex cover number used in previous analogous results for the strict variant). Finaly they show W[1]-hardness w.rt to other parameters.

**Strengths:**

Generalization to the non-strict model and strenthening of some hardness and FPT results.
The complexity picture of the problem is neater than before.

**Weaknesses:**

The gap in the upper and lower bounds are significant. I do not think that is a strong result.

**Questions:**

Are there results or ideas about approximation algorithms? Or hardness of approximation?

---

> ### Author Response · Authors · 2024-11-19
>
> Thank you for the positive review and nice question. We believe that approximation algorithms are an interesting next direction to take. To the best of our knowledge, no approximation algorithms are known. For the strict case, Theorem 1 in our paper shows that, unless P=NP, the problem cannot be 1.499-approximated in polynomial time. For the non-strict case, it can be inferred from (Kirkpatrick et al., 2019) that, unless P=NP, the problem cannot be 1.999-approximated in polynomial time. In the updated version of the submission, we have added this information in the related work and at the beginning of Section 3, and we have also added approximation algorithms as a future direction in the Concluding Remarks.

---

### Official Review · Reviewer_iU1d · 2024-11-06

**Soundness:** 4
**Presentation:** 3
**Contribution:** 3
**Rating:** 6
**Confidence:** 3

**Summary:**

The paper studies the complexity of computing the positive non-clashing teaching dimension that can be interpreted as the smallest number of examples needed to teach an intelligent learner. The paper models the concept class as the set of all balls of a certain radius k in a graph. (A ball of radius k consists of all vertices at a distance at most k from a vertex v serving as the center of the ball.)

The paper shows that the problem of computing the positive non-clashing teaching dimension is NP-complete, even when k=2. Previously, the problem was known to be NP-complete in general but not for a constant k.

The second result are improved upper and lower bounds on the time to solve the problem, as a function of the number of vertices n, diameter of the graph d and the radius of the balls k. Both previous and new bounds are exponential time but the exponents have been improved. For example, the upper bound is improved from 2^{O(n^2 d)} to 2^{O(n d k log n)}.

The third set of results is about the fixed parameter tractability of the problem. Previous, it was known that the problem is tractable if the vertex cover number of the graph is constant. This paper improves this to the vertex integrity being constant. It also considers several other possible improvements (replacing vertex integrity by smaller graph parameters) and rules them out.

**Strengths:**

Clear improvement over previous work.
Comprehensive set of results, covering the complexity of problem from various angles.
Original technical ideas.

**Weaknesses:**

Not sure about the fit to ICLR in terms of topic, as the paper proves general complexity-theoretic results but does not consider specific classes of concepts for which the studied questions have interesting implications.

**Questions:**

What would be concrete consequences of your results that would be interesting to ICLR audience?

---

> ### Author Response · Authors · 2024-11-19
>
> Thank you for your positive and constructive feedback. The concept class of balls in a graph for different graph classes is a well-motivated and natural concept class that has already been established in the literature. We believe that you are right in pointing out that we did not make this sufficiently clear. Thus, in the updated version of the submission, we have expanded the related work section (including adding a new paragraph) to address this.
>
> Concerning specific classes of concepts, in Theorem 1 we could also reduce from an NP-hard variant of 3-SAT in which each variable appears in at most a constant number of clauses, which would then make our hardness result for the strict version also apply in the case of concept classes of constant VC-dimension. As we also added before the statement of Theorem 1, our reduction implies that there is no 1.499-approximation algorithm unless P=NP. While both results could already be inferred from (Kirkpatrick et al., 2019) for the non-strict version (even no 1.999-approximation algorithm), our results hold for the strict version which defines a more natural concept class (a “full” discrete geometric class) admitting these hardness properties.

---

### Meta-Review · Area_Chair_3RPk · 2024-12-14

**Metareview:**

## Summary of Contributions

This paper studies the notion of *non-clashing teaching dimension*. The setting is that there is an undirected graph where each node represents an example and each concept is a ball in the graph. There are two subsettings: *strict* if all balls are valid concepts and *non-strict* otherwise. The goal is to find a teaching set (i.e. a subset) for each concept such that these teach sets can distinguish any pair of concepts. The main results of this paper are that the problem in NP-hard even in the strict setting and when the true non-clashing teaching dimension is only 2 (which is as good as possible as the 1 case is easy to solve). Additionally, the authors also give a slightly improve algorithm and show that their reduction gives a nearly matching running time lower bound, under the exponential-time hypothesis (ETH). They also study fixed parameter tractability of the problem under different parameters such as vertex integrity and feedback vertex number.

## Strengths

- Non-clashing teaching dimension is a well motivated concept that is well studied in literature, with connections to sample compression schemes. Thus, its computational complexity is important.

- This paper achieves strong lower bounds, both in terms of NP-hardness and running time lower bound, that are nearly optimal and, therefore, gives a nearly complete picture for the problem. The results also answer known open questions (Chalopin et al., COLT 2023).

## Weaknesses

- The results, proofs and the writing are quite specific. They might only appeal to those from the learning theory subcommunity and not the wider audience at ICLR.

## Recommendation

Given that the non-clashing teaching dimension is natural & well-studied and this paper makes a significant progress towards understanding its computational complexity, we recommend acceptance.

**Additional Comments On Reviewer Discussion:**

There were questions regarding the importance of non-clashing teaching dimension and approximation algorithms. For the former, the authors clarified by pointing out its connection to sample compression schemes and add that to a related work section. For the latter, the authors point out that their hardness already gives hardness of approximation of a factor ~1.5 already, so there can't be e.g. a polynomial-time approximation scheme (PTAS) for the problem.

---

### Decision · Program_Chairs · 2025-01-22

Accept (Poster)